



# Mechanistic Study of Formation of Ring-retaining and Ring-opening Products from Oxidation of Aromatic Compounds under Urban Atmospheric Conditions

Alexander Zaytsev[1], Abigail R. Koss[2], Martin Breitenlechner[1], Jordan E. Krechmer[3], Kevin J. Nihill[2], Christopher Y. Lim[2], James C. Rowe[2], Joshua L. Cox[4], Joshua Moss[2], Joseph R. Roscioli[3], Manjula R. Canagaratna[3], Douglas R. Worsnop[3], Jesse H. Kroll[2], and Frank N. Keutsch[1,4,5]

[1]John A. Paulson School of Engineering and Applied Sciences, Harvard University, Cambridge, MA 02138, USA,
[2]Department of Civil and Environmental Engineering, Massachusetts Institute of Technology, Cambridge, MA 02139, USA,
[3]Aerodyne Research Inc., Billerica, MA 01821, USA,
[4]Department of Chemistry and Chemical Biology, Harvard University, Cambridge, MA 02138, USA,
[5]Department of Earth and Planetary Sciences, Harvard University, Cambridge, MA 02138, USA

*Correspondence to*: Alexander Zaytsev (zaytsev@g.harvard.edu) and Frank N. Keutsch (keutsch@seas.harvard.edu)

**Abstract.** Aromatic hydrocarbons make up a large fraction of anthropogenic volatile organic compounds and contribute significantly to the production of tropospheric ozone and secondary organic aerosol (SOA). A series of toluene and 1,2,4-trimethylbenzene (1,2,4-TMB) photooxidation experiments were performed in an environmental chamber under relevant polluted conditions ($NO_x$ ~ 10 ppb). An extensive suite of instrumentation including two Proton-Transfer Reaction Mass-Spectrometers (PTR-MS) and two Chemical Ionization Mass-Spectrometers ($NH_4^+$ CIMS and I⁻ CIMS) allowed for quantification of reactive carbon in multiple generations of oxidation. Hydroxyl radical (OH)-initiated oxidation of both species produces ring-retaining products such as cresols, benzaldehydes, and bicyclic intermediate compounds, as well as ring scission products such as epoxides, and dicarbonyls. We show that the oxidation of bicyclic intermediate products leads to formation of compounds with high oxygen content (O:C ratio up to 1.1). These compounds, previously identified as highly oxygenated molecules (HOMs), are produced by more than one pathway with differing numbers of reaction steps with OH, including both autooxidation and phenolic pathways. We report the elemental composition of these compounds formed under relevant urban high-NO conditions. We show that ring-retaining products for these two precursors are more diverse and abundant than predicted by current mechanisms. We present speciated elemental composition of SOA for both precursors and confirm that highly oxygenated products make up a significant fraction of SOA. Ring scission products are also detected in both the gas and particle phases, and their yields and speciation overall agree with the kinetic model prediction.



## 1 Introduction

Aromatic compounds represent a significant fraction of volatile organic compounds (VOCs) in the urban atmosphere and play a substantial role in the formation of tropospheric ozone and secondary organic aerosol (SOA) (Calvert et al., 2002). Typical anthropogenic sources include vehicle exhaust, solvent use, and evaporation of gasoline and diesel fuels. Toluene, the most

abundant alkylbenzene in the atmosphere, is primarily emitted by aforementioned anthropogenic processes. Toluene-derived SOA is estimated to contribute approximately 17-29% of the total SOA produced in urban areas (Hu et al., 2008). More highly substituted aromatic compounds make up another important group of aromatic compounds as they tend to have high SOA yields (Li et al., 2016) and account for a significant fraction of non-methane hydrocarbons in the industrialized regions of China (Tang et al., 2007; Zheng et al., 2009). 1,2,4-trimethylbenzene (1,2,4-TMB) serves as a model molecule to study

oxidation of more substituted aromatic compounds.

In the atmosphere, oxidation of aromatic hydrocarbons is most often initiated by their reactions with hydroxyl radicals (OH) via H-abstraction from the alkyl groups or OH addition to the aromatic ring (Fig. 1) (Calvert et al., 2002). The abstraction channels are relatively minor, leading to the formation of benzyl radicals and benzaldehyde with yields of ~0.07 in the case of toluene (Wu et al., 2014) and ~0.06 in the case of 1,2,4-TMB (Li and Wang, 2014). The OH-adducts can react with atmospheric

$O_2$ through H-abstraction to form ring-retaining phenolic compounds (i.e., cresols and trimethylphenols). The phenol formation yield decreases for the more substituted aromatics: in case of toluene the cresol yield is ~0.18 (Klotz et al., 1998, Smith et al., 1998) while for 1,2,4-TMB the trimethylphenol yield is ~0.03 (Bloss et al., 2005). Both abstraction and phenolic channels lead to the formation of products retaining the aromatic ring.

The OH-adducts can also react with $O_2$ through recombination. In this case they lose aromaticity and form non-aromatic ring-

retaining bicyclic peroxy radicals (BPRs). Under low-NO conditions, BPRs react with $HO_2$ and $RO_2$, forming bicyclic hydroperoxides and bicyclic carbonyls, respectively (Fig. 2). In addition, BPRs can undergo unimolecular H-migration followed by $O_2$-addition (so-called autooxidation) leading to the formation of non-aromatic ring-retaining highly oxygenated organic molecules (HOMs) (Bianchi et al., 2019). Molteni et al. (2018) reported elemental composition of the HOMs from a series of aromatic compounds produced under low-NO conditions. The autooxidation pathway might be more important for

the substituted aromatics because of the higher yield of BPR formation and the larger number of relatively weak C-H bonds (Wang et al., 2017). Under urban-relevant high-NO conditions BPRs also react with NO to form bicyclic oxy radicals that decompose to ring scission carbonylic products such as (methyl) glyoxal and biacetyl. Recent theoretical studies predict a new type of epoxy-dicarbonyl products that have not reported in previous studies (Li and Wang, 2014, Wu et al., 2014). Reaction of BPRs with NO can also result in formation of bicyclic organonitrates. Both ring-retaining and ring scission compounds are

expected to be low in volatility and contribute significantly to SOA (Schwantes et al., 2017). There remain a number of major uncertainties in model representation of oxidation of aromatic compounds including overprediction of ozone concentration, underprediction of OH production and lack of detailed description of SOA formation (Birdsall and Elrod, 2011; Wyche et al., 2009).





In the present work, we investigate the detailed mechanism of hydroxyl radical multigeneration oxidation chemistry of two aromatic hydrocarbons: toluene and 1,2,4-trimethylbenzene under moderate, urban-relevant $NO_x$ levels (~10 ppbv). Laboratory experiments were conducted at an environmental chamber over approximately 1 day of atmospheric-equivalent oxidation. We use four high-resolution time-of-flight chemical ionization mass spectrometers ($NH_4^+$ CIMS, $I^-$ CIMS and two PTR-MS) to characterize and quantify gas- and particle-phase oxidation products. We identify gas-phase pathways leading to production of low-volatility compounds which are important for SOA formation and support these identifications with CIMS data and a method to characterize the kinetics of an oxidation system.

## 2 Methods

### 2.1 Experimental design

All experiments were performed in a 7.5 m$^3$ Teflon environmental chamber (Hunter et al., 2014). Prior to experiments, the chamber was flushed and filled with purified air. During experiments clean air was continuously added to the chamber to keep its volume constant. The temperature of the chamber was controlled at 292 K and approximately 2% relative humidity.

We performed a series of photochemical experiments, in which toluene and 1,2,4-TMB were oxidized by OH under high-NO conditions (Table S1). First, dry ammonium sulfate particles, used as condensation nuclei, were injected in the chamber to reach a number concentration of 2.5-5.7·10$^4$ cm$^{-3}$. Seed particles were not injected into the chamber in two experiments. Hexafluorobenzene, ($C_6F_6$, which serves as a dilution tracer) was then added to the chamber. Nitrous acid (HONO) was later injected as an OH precursor. HONO was generated in a bubbler containing a solution of sodium nitrite by adding 2-4 $\mu$L of sulfuric acid via a syringe pump. 15 lpm of subsequently injected purified air carried HONO into the chamber, which resulted in a concentration of 35-45 ppbv. The concentration of NO in the chamber was estimated to be ~0.3 ppbv while $NO_2$ concentration was approximately 10 ppbv. After the addition of the oxidant, the aromatic precursor (toluene or 1,2,4-TMB, Sigma-Aldrich) was added to the chamber by injecting 3 $\mu$L of the precursor into a heated inlet. The initial concentration of the precursor was 89 ppbv in toluene experiments and 69 ppbv in 1,2,4-TMB experiments. The reagents were allowed to mix for several minutes, after which the ultraviolet (UV) lights, centred ~340 nm, were turned on to start photolysis of HONO and photooxidation of the precursor. During experiments, additional injections of HONO were added to the chamber in order to roughly maintain the OH levels. Measurements were conducted within several hours, which corresponds to 14-16 hours of atmospheric-equivalent exposure (assuming an average OH concentration of 1.5·10$^6$ molecules cm$^{-3}$). Concentrations of $O_3$ and HONO+$NO_x$ for a typical run are shown on Fig S1.

### 2.2 Chamber instrumentation

The concentration of nitrogen oxides ($NO_x$) and HONO (42i $NO_x$ monitor, Thermo Fisher Scientific), ozone (2B Technologies), relative humidity, and temperature were measured in the chamber. Aromatic precursors as well as gas-phase oxygenated volatile organic compounds (OVOCs) were detected by chemical ionization high-resolution time-of-flight mass



spectrometry (CIMS) instruments, including the I⁻ CIMS instrument (Aerodyne Research Inc.; Lee et al., 2014) and two proton-transfer-reaction mass-spectrometry (PTR-MS) instruments: Vocus-2R-PTR-TOF (TOFWERK A.G.; Krechmer et al., 2018) and PTR3 (Ionicon Analytik; Breitenlechner et al., 2017). The latter instrument was operated in a switching-mode regime using $H_3O^+ \cdot (H_2O)_n$, n=0-1 (as $H_3O^+$ CIMS) and $NH_4^+ \cdot (H_2O)_n$, n = 0-2 (as $NH_4^+$ CIMS) primary ions (Hansel et al., 2018;

Zaytsev et al., 2019). Switching between ion modes occurred every five minutes. Each CIMS instrument used a 3/16'' PFA Teflon sampling line of 1 m in length with a flow of 2 slm. PTR3 and Vocus-2R-PTR-TOF are designed to minimize inlet losses of sampled compounds (Krechmer et al., 2018; Breitenlechner et al., 2017). Detection efficiency and sensitivity of CIMS instruments depend critically on both the reagent ion and the measured sample molecule. The concentrations of aromatic precursors were measured by Vocus-2R-PTR-TOF, which was directly calibrated for the two compounds. Smaller, less

oxidized molecules were primarily quantified by PTR-MS (PTR3 $H_3O^+$ CIMS and Vocus-2R-PTR-TOF) while PTR3 $NH_4^+$ CIMS and I⁻ CIMS were mainly used for detection of larger and more functionalized molecules. PTR3 and I⁻ CIMS instruments were directly calibrated for 10 VOCs with various functional groups. An average calibration factor was applied to other species detected by PTR-MS instruments, while collision-dissociation methods were implemented to constrain sensitivities of I⁻ CIMS and $NH_4^+$ CIMS (Lopez-Hilfiker et al., 2016; Zaytsev et al., 2019). Use of these methods and average calibration factors leads

to uncertainties in estimated concentrations of detected compounds within a factor of ten. $NH_4^+$ CIMS uncertainties are within a factor of three while I⁻ CIMS is less certain. The majority of analysis in this work relies on $NH_4^+$ CIMS and PTR-MS data while I⁻ CIMS data are used as supporting measurements. CO and formaldehyde were measured by tunable infrared laser differential absorption spectroscopy (TILDAS; Aerodyne Research Inc.), and glyoxal was detected by laser induced phosphorescence (Madison LIP; Huisman et al., 2008).

Total organic aerosol mass was measured using an Aerodyne Aerosol Mass Spectrometer (AMS, DeCarlo et al., 2006), calibrated with ammonium nitrate and assuming a collection efficiency of 1. Particle-phase compounds were quantified using the FIGAERO-HRToF-I⁻ CIMS (Lopez-Hilfiker et al., 2014), and a second PTR3 that could be operated in two positive modes as described above and equipped with an aerosol inlet comprising a gas-phase denuder and a thermal desorption unit (TD-$NH_4^+$ CIMS and TD-$H_3O^+$ CIMS) (Zaytsev et al., 2019).

**2.3 Kinetic model**

The Framework for 0-D Atmospheric Modelling v3.1 (F0AM; Wolfe et al., 2016) containing reactions from the Master Chemical Mechanism (MCM v3.3.1) (Jenkin et al., 2003; Bloss et al., 2005) was used to simulate photooxidation of 1,2,4-TMB and toluene in the environmental chamber and to compare the modelled products with the measurements. Model calculations were constrained to physical parameters of the environmental chamber (pressure, temperature, photolysis

frequencies and dilution rate calculated using the hexafluorobenzene tracer). Injections of aromatic compounds and HONO were modelled as sources during the time of injection, and the chamber lights intensity was tuned to match the measured time-dependent concentration of aromatic precursors with the modelled values. The chamber wall loss and dilution term for volatile compounds was estimated based on the concentration of the dilution tracer, hexafluorobenzene. As for semi- and low-volatile



compounds, the wall deposition rate was estimated to be $5 \cdot 10^{-4}$ s$^{-1}$ using the "rapid burst" method described in detail by Krechmer et al. (2016).

**2.4 Calculation of OH exposure and product yields**

The OH concentration was determined using the decay of the aromatic precursor, accounting for losses from dilution and chamber wall deposition. The mixing ratio of the aromatic VOC (ArVOC: toluene or 1,2,4-TMB) is given by the following kinetics equation:

$$[\text{ArVOC}]_t = [\text{ArVOC}]_0 \cdot \exp(- k_{\text{ArVOC+OH}} \cdot [\text{OH}_{\text{exposure}}]_t - k_{\text{physical loss ArVOC}} \cdot t) \tag{1}$$

where [ArVOC] is the time-dependent mixing ratio of the aromatic precursor, $k_{\text{ArVOC+OH}}$ is the second-order rate constant for ArVOC + OH reaction ($k_{\text{toluene+OH}} = 5.63 \cdot 10^{-12}$ cm$^3$ molecule$^{-1}$ s$^{-1}$ and $k_{1,2,4-\text{TMB+OH}} = 3.25 \cdot 10^{-11}$ cm$^3$ molecule$^{-1}$ s$^{-1}$ at 293K (Calvert et al., 2002)), $[\text{OH}_{\text{exposure}}]_t = \int_o^t [\text{OH}] d\tau$ is the integrated OH exposure, $k_{\text{physical loss ArVOC}}$ is the unimolecular rate constant determining dilution and chamber wall loss, and $t$ is the time since the beginning of irradiation by the UV lights.

Yields of first-generation products were determined based on the decay of the aromatic precursor, rise of the product, and accounting for physical (dilution and chamber wall deposition) and chemical losses (reaction with OH, NO$_3$, O$_3$ and photolysis). A correction procedure described in detail by Galloway et al. (2011) is applied to calculate product yields. This correction takes into account physical and chemical losses of products in the environmental chamber:

$$[\text{X}]_i^{\text{corr}} = [\text{X}]_{i-1}^{\text{corr}} + \Delta[\text{X}]_i + [\text{X}]_{i-1} \Delta t (k_{\text{chemical loss}} + k_{\text{physical loss}}) \tag{2}$$

where $[\text{X}]_i^{\text{corr}}$ is the corrected mixing ratio of the compound at measurement time $i$, $\Delta t$ is time between measurements $i$ and $i-1$, $[\text{X}]_{i-1}$ is the measured product mixing ratio of measurement $i-1$, $\Delta[\text{X}]_i$ is the observed net change in [X] that occurs over $\Delta t$, $k_{\text{chemical loss}}$ and $k_{\text{physical loss}}$ are the rate constants describing chemical and losses of the product.

Yields of first-generation products were determined from the linear relationship between the amount of the corrected product formed and the amount of the primary ArVOC reacted:

$$[\text{X}]^{\text{corr}} = Y[\text{ArVOC}]^{\text{reacted}} + b \tag{3}$$

where $[\text{X}]^{\text{corr}}$ is the amount of the corrected product formed, $[\text{ArVOC}]^{\text{reacted}}$ is the amount of the primary ArVOC reacted and $Y$ is the first-generation yield of the product.

**2.5 Gamma kinetics parameterization**

Photooxidation products can be characterized not only by their concentration and yield, but also by their time series behaviour. In a laboratory experiment, the time series behaviour of a product is dependent on the kinetic parameters of the molecule: the relative rates of formation and reactive loss, and the number of reactions to create the product. We characterize time series behaviour of products using the gamma kinetics parametrization (GKP), which describes kinetics of an oxidation system in terms of multigenerational chemistry. The detailed description of this parametrization technique is given by Koss et al. (2019),





so we include only a brief description here. A multigenerational hydroxyl radical oxidation system can be represented as a linear system of reactions:

$$X_0 \xrightarrow{k\cdot[\text{OH}]} X_1 \xrightarrow{k\cdot[\text{OH}]} X_2 \xrightarrow{k\cdot[\text{OH}]} ... \xrightarrow{k\cdot[\text{OH}]} X_m \xrightarrow{k\cdot[\text{OH}]} X_{m+1} \xrightarrow{k\cdot[\text{OH}]} ... \tag{4}$$

where $k$ is the second-order rate constant and $m$ is the number of reactions needed to produce species $X_m$.

In laboratory experiments, oxidation reactions can be parametrized as a linear system of first-order reactions if reaction time $t$ is replaced by OH exposure $[\text{OH}_{\text{exposure}}]_t = \int_0^t [\text{OH}]d\tau$. In this case, the time-dependent concentration of a compound $X_m$ can be parametrized by (Koss et al., 2019):

$$[X_m](t) = a(k \cdot [\text{OH}_{\text{exposure}}]_t)^m e^{-k\cdot[\text{OH}_{\text{exposure}}]_t} \tag{5}$$

where $a$ is a scaling factor, $k$ is the effective second-order rate constant (cm$^3$ molecule$^{-1}$ s$^{-1}$), $m$ is the generation number, and $[\text{OH}_{\text{exposure}}]_t$ is the integrated OH exposure (molecule s cm$^{-3}$).

Eq. (5) can be used to fit the observed concentration of a compound to return its parameters $a$, $k$ and $m$. The parameter $m$ determines the number of reactions with OH needed to produce the compound while the parameter $k$ gives an approximate measure of the compound reactivity. We define "generation" here as the number of reactions with OH. Examples of fitted chamber measurements are shown in Fig. S2.

## 3 Results and discussion

Toluene and 1,2,4-TMB react with OH to form both ring-retaining (via benzaldehyde, phenolic and bicyclic channels) and ring scission (via bicyclic and epoxide channels) products. (The toluene oxidation scheme from MCM v3.3.1 is shown on Fig. 1.) First and later-generation gas- and particle-phase oxidation products are detected and quantified for both systems. In the following sections we compare products suggested by MCM and previous studies to corresponding molecular formulas detected by CIMS. In some cases, the ion identity is well established from previous research or because there are a limited number of reasonable structures (e.g., phenols, benzaldehydes and ring-scission dicarbonyl products); in other cases, multiple isomers are possible, which could contribute to some differences between modelled and observed behaviour.

### 3.1 Products from benzaldehyde, phenolic and epoxide channels

In the toluene experiments, the approximate yields of benzaldehyde and cresol (~0.10 and ~0.16 correspondingly) were calculated based on the decay of toluene measured by Vocus-2R-PTR-TOF, rise of the two products measured by PTR3 $H_3O^+$ CIMS and $NH_4^+$ CIMS, and accounting for losses of cresol and benzaldehyde from wall deposition and reaction with OH and $NO_3$ (Sect 2.4). MCM v3.3.1 recommends a 0.07 yield of benzaldehyde and a 0.18 yield of cresol (total of all isomers) from OH initiated oxidation of toluene. Benzaldehyde and cresol concentrations predicted by MCM agree within uncertainties with the PTR3 $H_3O^+$ CIMS and $NH_4^+$ CIMS measurements and the time-series behaviour of measurements and model predictions are similar (Fig. 3a). In general, MCM predicts that phenolic and benzaldehyde channels are less important for more substituted



aromatics (Bloss et al., 2005). Hence, the kinetic model recommends a 0.06 yield of dimethylbenzaldehyde and a 0.03 yield of trimethylphenol from the OH-initiated oxidation of 1,2,4-TMB. The model predictions for the two products agree within uncertainties with the PTR-MS measurements and the time series behaviour is again similar (Fig. 3b). Phenols and benzaldehydes can further react within the MCM v3.3.1 scheme to form highly oxidized second-generation compounds. The

importance of this pathway is discussed further in section 3.2.1.

MCM v3.3.1 also includes an epoxy-oxy channel in which it predicts formation of non-fragmentary linear epoxide-containing products (Fig. 1). The predicted yields of these species are 0.10 and 0.30 for toluene and 1,2,4-TMB systems, respectively. The observed yields of these products are significantly smaller (~0.01 in both systems). The observations are however consistent with theoretical studies (Li and Wang, 2014; Wu et al., 2014) in which it has been shown that only a negligible

fraction of the bicyclic radicals would break the -O-O- bond to form epoxide-containing products.

## 3.2 Products from bicyclic pathway

### 3.2.1 Non-fragmented, ring-retaining products

Bicyclic peroxy radicals (BPRs) are formed through the addition of $O_2$ to the OH-adducts (Fig. 1). Starting from a generic aromatic compound $C_xH_y$, we expect the formation of BPRs with the formula $C_xH_{y+1}O_5$. BPRs likely react with $RO_2$, $HO_2$, or

NO leading to the following products (Fig. 2) (Birdsall and Elrod, 2011): bicyclic carbonyls ($C_xH_yO_4$), bicyclic alcohols ($C_xH_{y+2}O_4$), bicyclic hydroperoxides ($C_xH_{y+2}O_5$), and bicyclic organonitrates ($C_xH_{y+1}NO_6$), as well as alkoxy radicals (discussed later). A number of oxygenated products is detected by $NH_4^+$ CIMS including $C_7H_8O_4$, $C_7H_{10}O_5$ and $C_7H_9NO_6$ in the toluene experiments, and $C_9H_{12}O_4$, $C_9H_{14}O_4$ and $C_9H_{13}NO_6$ in the 1,2,4-TMB experiments (Table 3). Since authentic standards are not available, quantification of these compounds was done using a voltage scanning procedure based on collision-induced

dissociation (Sect. 2.2). The majority of OVOCs with high carbon numbers were detected at the maximum sensitivity, which was experimentally determined in each photooxidation experiment and depends on operational conditions of the $NH_4^+$ CIMS instrument (Zaytsev et al., 2019) (Tables S2 and S3). A number of products with the same molecular formulas corresponding to bicyclic carbonyls ($C_7H_8O_4$ and $C_9H_{12}O_4$) and alcohols ($C_7H_{10}O_4$ and $C_9H_{14}O_4$) were also detected by I$^-$ CIMS and PTR-MS.

In MCM v3.3.1, the bicyclic peroxy radical undergoes analogous reactions: (1) it can react with $HO_2$ producing a hydroperoxide; (2) it can react with $RO_2$ producing an alkoxy radical, an alcohol or a carbonyl; (3) it can react with NO producing an alkoxy radical or a nitrate. According to MCM, under relevant urban conditions BPRs nearly exclusively react with NO and $HO_2$ and dominantly form alkoxy radicals which further decompose to smaller ring scission compounds (Figs. 2, S3 and S4) (Sect. 3.2.3). In this study, the estimated lifetime of BPRs, calculated as inverse reactivity, is estimated to be ~5

30   s for toluene and ~7 s for 1,2,4-TMB. MCM v3.3.1 does not include formation of the bicyclic carbonyl, $C_9H_{12}O_4$, in the 1,2,4-TMB oxidation scheme, while in the case of toluene it predicts that a major fraction of bicyclic carbonyl $C_7H_8O_4$ is produced as a second-generation product from the reaction of bicyclic hydroperoxide and organonitrate with OH. In contrast, the GKP





fit based on the $NH_4^+$ CIMS measurements implies that a significant fraction of detected compounds is formed in the first generation in both systems (Table 3). These first-generation bicyclic carbonyls can be produced by the reaction of BPR with $HO_2$ or $RO_2$. In addition, MCM significantly underestimates the concentration of bicyclic alcohol ($C_9H_{14}O_4$ for 1,2,4-TMB system) since the only channel present in the model is a reaction of BPR with $RO_2$ while it can also be produced via the

5 BPR+$HO_2$ pathway (Fig. 2). Finally, MCM v3.3.1 predicts that a notable fraction of BPR reacts with $HO_2$ to form bicyclic hydroperoxide. $NH_4^+$ CIMS measurements of $C_9H_{14}O_5$ and $C_7H_{10}O_5$ are less than the model prediction for each chemical system but agree within measurement uncertainties. The generation number $m$ of $C_7H_{10}O_5$ and $C_9H_{14}O_5$ is ~1.7-1.8 which suggests that a compound with the same molecular formula can be produced by more than one pathway with different number of reaction steps. Birdsall and Elrod (2011) proposed that the BPR from several aromatic precursors reacting with $HO_2$ can form an alkoxy

radical and OH. Similarly, recent studies have shown that numerous peroxy radicals do not form a hydroperoxide in unity yield while reacting with $HO_2$ (Praske et al., 2015; Orlando and Tyndall, 2012). Hence, we observe formation of numerous highly oxygenated compounds with molecular formula corresponding to bicyclic carbonyls, alcohols, organonitrates, and hydroperoxides via the peroxide-bicyclic channel under high-NO conditions.

In addition to the formation of closed-shell products, bicyclic peroxy radicals can undergo isomerization reactions to form

more oxidized peroxy radicals (Fig. 2) (Wang et al., 2017; Molteni et al., 2018). These reactions are not included in MCM v3.3.1. These radicals can in turn react with $RO_2$, $HO_2$, or NO leading to a series of more oxidized bicyclic products, so-called highly oxygenated molecules (HOMs): carbonyl ($C_xH_yO_6$), alcohol ($C_xH_{y+2}O_6$), hydroperoxide ($C_xH_{y+2}O_7$), and nitrate ($C_xH_{y+1}NO_8$). Several products with the aforementioned molecular formulas are detected by $NH_4^+$ CIMS (Fig. 4). In the toluene experiments, $C_7H_8O_6$, $C_7H_{10}O_6$, and $C_7H_9NO_8$ were detected by $NH_4^+$ CIMS as $(NH_4^+)\cdot OVOC$. In the 1,2,4-TMB experiments,

$C_9H_{14}O_6$, and $C_9H_{13}NO_8$ were detected as $(NH_4^+)\cdot OVOC$ while $C_9H_{12}O_6$ was detected both as $(NH_4^+)\cdot C_9H_{12}O_6$ (m/z 234.098) and as $[(NH_4)\cdot(H_2O)\cdot C_9H_{12}O_6]^+$ (m/z 252.108). The yield of molecules with a high O:C (greater than 0.44) ratio is larger in the case of 1,2,4-TMB compared to toluene. However, the gamma kinetics parametrization suggests that none of these compounds are solely first-generation products (Table 3) which implies that there are multiple chemical pathways in which products with the same molecular formula (but potentially different structure) are formed. Some of the products can be produced in the

bicyclic oxidation pathway of cresol or trimethylphenol resulting in formation of alcohols ($C_xH_{y+2}O_5$), hydroperoxides ($C_xH_{y+2}O_6$) and nitrates ($C_xH_{y+1}NO_7$). Formation of bicyclic alcohols and hydroperoxides from the phenolic channel is included in MCM v3.3.1. We conclude that although a plethora of compounds with a high O:C ratio was observed, only some of them are formed via first-generation isomerization/autoxidation pathways. These findings underline the importance of phenolic and benzaldehyde channels for producing highly oxygenated compounds especially for the less substituted aromatics given

the high yields of phenols and benzaldehydes and their high reactivity.

### 3.2.2 Fragmented ring-retaining products

In addition to non-fragmented functionalized $C_9$ and $C_7$ products possibly formed via the bicyclic channel, a variety of lower-carbon containing ($C_8$ and $C_6$) products with fewer carbon atoms were also detected in both chemical systems. The total



concentration of $C_8$ components predicted by MCM v3.3.1 for 1,2,4-TMB is in good agreement with the $NH_4^+$ CIMS and PTR-MS measurements, though the observed composition is distinctly different from the MCM prediction (Fig. 5). According to MCM, there are only four $C_8$ products with mixing ratios at or above the ppt level. The most abundant predicted compound, bicyclic carbonyl ($C_8H_{10}O_4$, MXYOBPEROH in MCM), is produced via the reaction of bicyclic nitrate $C_9H_{13}NO_6$ with OH, while the second most abundant compound, dimethylnitrophenol ($C_8H_9NO_3$, DM124OHNO2 in MCM), is formed in the benzaldehyde pathway. While all of the predicted products have been observed, $NH_4^+$ CIMS and PTR-MS also detect a plethora of $C_8$ compounds not included in MCM. Furthermore, MCM only predicts a total of 0.16 ppb for all $C_6$ compounds in the toluene experiments, while the combined measured mixing ratio of 15 $C_6$ compounds is ~0.5 ppb (Fig S5). The two most prominent $C_6$ products recommended by MCM, $C_6H_5NO_3$ and $C_6H_6O_2$, are predicted to be formed in the benzaldehyde channel. There are several pathways that may lead to the formation of the fragmented ring-retaining $C_8$ and $C_6$ products. One of them is ipso addition followed by dealkylation (Loison et al., 2012). In their study of the dealkylation pathway in the OH-initiated oxidation of several aromatic compounds, Noda et al. (2009) showed the importance of this pathway and reported dealkylation yields of 0.054 for toluene. Similarly, Birdsall and Elrod determined a 0.047 yield of dealkylation products for toluene. Loison et al. (2012) reported a 0.02 yield of dealkylation pathway products from the hexamethylbenzene + OH reaction. However, in some studies yields of dealkylation products have been found below 0.01 (Aschmann et al., 2010). While we cannot determine exact yields of observed dealkylation compounds since many of them are later-generation products, we estimate an overall amount of carbon of ~3 ppbC and ~5 ppbC stored in $C_6$ and $C_8$ compounds for toluene and 1,2,4-TMB, respectively, compared to 600 ppbC in the system as a whole. Although observed concentrations of these products are not very high, a significant fraction of them are highly oxygenated products with an O:C ratio greater than 0.5. These products can effectively partition to the particle phase and contribute to SOA formation (Section 3.3).

### 3.2.3 Products from ring scission pathway

Recent theoretical studies suggest that the ring scission pathway leads to a series of two ring scission products such that the total number of carbon atoms present in the original aromatic compound is conserved in the two products (Wu et al., 2014; Li and Wang, 2014). In particular, a toluene study by Wu et al. (2014) predicts significant yields of 1,2-dicarbonyls and a range of $C_4$ and $C_5$ products. Those larger compounds include dicarbonyls (butenedial $C_4H_4O_2$ and methylbutenedial $C_5H_6O_2$), which were already detected in previous experimental studies (Calvert et al., 2002), and newly proposed epoxy-dicarbonyl products (epoxybutanedial $C_4H_4O_3$ and methylepoxybutanedial $C_5H_6O_3$). As for 1,2,4-TMB, Li and Wang (2014) also predict two groups of ring scission products: (1) smaller 1,2-dicarbonyls such as glyoxal, methylglyoxal and biacetyl and (2) larger $C_5$, $C_6$ and $C_7$ products. Similar to the toluene system, those larger compounds include dicarbonyls ($C_5H_6O_2$, $C_6H_8O_2$ and $C_7H_{10}O_2$), which were previously reported in numerous studies, and new epoxy-dicarbonyl products ($C_5H_6O_3$ and $C_6H_8O_3$). Some experimental studies reported that larger ring scission products were found at systematically lower yields than the corresponding 1,2-dicarbonyl products (Arey et al., 2009). This result suggests that either the product pair carbon conservation





rule is not followed or that the larger products undergo further photochemical or heterogeneous degradation in the environmental chamber.

A series of 1,2-dicarbonyls were experimentally detected including glyoxal and methylglyoxal in the toluene experiments, and methylglyoxal and biacetyl in the 1,2,4-TMB experiments (Tables 1 and 2). Although a relatively small amount of glyoxal

(2.5 ppb) is predicted to be formed from the 1,2,4-TMB + OH pathway, observed amounts of glyoxal were below the limit of detection of the Madison LIP instrument (1 ppb) suggesting that the glyoxal yield does not exceed 0.03 in this system. Measured glyoxal and methylglyoxal formation yields, combined with the biacetyl yield from the 1,2,4-TMB oxidation, indicate that under high-NO conditions the first-generation yield of all 1,2-dicarbonyls is ~0.43 and ~0.35 for toluene and 1,2,4-TMB, respectively (Tables 1 and 2). For all 1,2-dicarbonyls the observed generation number $m$ is greater than 1 and is

consistent between the two systems, which suggests that these compounds are produced by more than one pathway with different numbers of reaction steps. In addition to the aforementioned 1,2-dicarbonyls, a series of dicarbonyls with a higher carbon number was observed in both systems (Tables 1 and 2). For all these species (except trimethylbutenedial) the generation number $m$ is slightly smaller than 1. Finally, ions corresponding to newly proposed epoxy-dicarbonyl products were also observed (Tables 1 and 2). However, the generation number $m$ for these compounds is between 1 and 2, suggesting that there

are multiple pathways resulting in compounds with those molecular formulas. Overall, we observed ~0.45 and ~0.47 yields of larger carbonyls in the toluene and 1,2,4-TMB experiments, respectively.

### 3.3 SOA Analysis

Peak SOA concentration measured by TD-NH$_4^+$ CIMS was 1.8 $\mu$g m$^{-3}$ for the toluene oxidation experiment and 1.4 $\mu$g m$^{-3}$ for 1,2,4-TMB which is in good agreement with the AMS measurements (1.9 $\mu$g m$^{-3}$ and 1.6 $\mu$g m$^{-3}$, respectively). Figure 6 depicts

the relative distribution of carbon in the particle phase according to the carbon atom number $n_C$ for 1,2,4-TMB and toluene SOA, respectively. The O:C ratios calculated from individual species measured from thermally desorbed SOA using NH$_4^+$ CIMS were ~0.95 for toluene SOA and ~0.7 for 1,2,4-TMB SOA. These ratios are in good agreement with the atomic O:C ratios measured by AMS (0.85 and 0.65 for toluene and 1,2,4-TMB SOA, respectively) (Canagaratna et al., 2015).

Products observed in the gas phase are compared to those detected in the particle phase to further understand the mechanism

of SOA formation from aromatics precursors. A variety of ring-retaining products were observed in the particle phase for oxidation of 1,2,4-TMB and toluene: non-fragmentary products (e.g., $C_9H_{12}O_{4-6}$ for 1,2,4-TMB and $C_7H_8O_{4-6}$ for toluene) and fragmentary products (e.g., $C_8H_{10}O_{4-5}$, $C_7H_8O_{4-5}$ for 1,2,4-TMB and $C_6H_6O_{4-5}$ for toluene). The same non-fragmentary ring-retaining oxidation products, detected by NH$_4^+$ CIMS in the gas phase and expected to be low in volatility, are detected in the particle phase (Fig. 4). Overall, ring-retaining products make up a significant fraction of the total 1,2,4-TMB SOA mass (~25%

for 1,2,4-TMB SOA and ~30% for toluene SOA) (Fig. 6), even though the total concentration of these products in the gas phase is relatively small (~2 ppb for 1,2,4-TMB and ~1 ppb for toluene). Furthermore, numerous masses corresponding to those of the ring scission products were detected including $C_5H_8O_3$, $C_4H_6O_3$, and $C_3H_4O_2$. It is likely that the concentration of larger ring-retaining products in the particle phase is underestimated due to their thermal fragmentation in the desorption unit



inlet of the $NH_4^+$ CIMS instrument (Zaytsev et al., 2019) although it should be noted that many highly oxygenated compounds were detected, e.g., $C_7H_6O_6$ and $C_9H_{10}O_7$. These findings underscore the importance of bicyclic, phenolic and benzaldehyde channels for producing ring-retaining highly oxygenated compounds that comprise a significant fraction of SOA (more than 25%).

## 4 Conclusions and implications for the aromatic oxidation mechanism

In the present work, we studied the multigenerational photooxidation of two aromatic compounds (toluene and 1,2,4-trimethylbenzene) in an environmental chamber under relevant urban high-NO conditions. We identified a number of oxidation products based on their molecular formula and determined yields for certain first-generation products. We provided kinetic and mechanistic information on numerous products using a gamma kinetics parameterization fit.

We compare and contrast observed products with predictions from the Master Chemical Mechanism (MCM v3.3.1), which provides explicit representation of chemical reactions that constitute the overall aromatic oxidation mechanism on the basis of the extensive body of existing experimental work. Laboratory studies are a vital support for the model, and the agreement between these studies and the MCM output is one of the important criteria demonstrating the accuracy of the kinetic model prediction. MCM accurately predicts the overall presence and importance of the three major channels for the primary OH-initiated oxidation of aromatic compounds (peroxide-bicyclic, benzaldehyde and phenolic). However, the epoxy-oxy channel appears to be overpredicted by the kinetic model. In both systems we observed a variety of stable bicyclic products which underlines the importance of the bicyclic channel in the oxidation of aromatic compounds (Fig. 1). MCM correctly predicts the greater importance of the bicyclic pathway for the more substituted aromatics, though the speciated fate of BPRs is not entirely consistent with observations. In addition to bicyclic organonitrates, carbonyls, and alcohols, we detect numerous compounds with molecular formulas corresponding to bicyclic hydroperoxides, which suggests that this class of products may be formed in the oxidation of aromatic molecules even under high-NO conditions. Recent studies (Molteni et al., 2018; Wang et al., 2017) suggest that bicyclic peroxy radicals can undergo isomerization and autooxidation reactions leading to the formation of HOMs. Furthermore, Molteni et al. (2018) report that the HOM yields from various aromatics are relatively uniform and are not linked to bicyclic peroxy radical or phenol formation yields. Here, we observe the formation of highly oxygenated compounds with a high O:C ratio (up to 1.1) in both systems. However, the abundance of these compounds is greater in the case of 1,2,4-TMB for which the yield of bicyclic peroxy radicals is higher (Fig. 4). Moreover, the observed kinetics of these compounds suggests that they might be produced by more than one pathway, including both the isomerization/autooxidation reactions and further oxidation of phenols and benzaldehydes. Many of these compounds are low in volatility and comprise a significant fraction (more than 25%) of SOA mass. These findings emphasize the significance of ring-retaining highly oxygenated products for SOA formation and provide further evidence that isomerization and autooxidation reactions can be fast enough to compete with bimolecular reactions with NO and $HO_2$ even under high-NO conditions (in this study the total BPR reactivity is estimated to be ~0.1-0.2 $s^{-1}$ (Figs. S3 and S4)). A plethora of fragmentary





ring-retaining products is observed in the gas phase in both systems, many of which are not present in MCM, which underlines the importance of the fragmentation pathway in oxidation of aromatic compounds. Numerous ring scission products are detected, including $C_2$-$C_7$ dicarbonyls and $C_{4-6}H_{4-8}O_3$ products, possibly epoxydicarbonyls. The kinetics of the ring scission products (e.g., generation number $m$) is consistent between the two systems, which suggests that many of those carbonyls are

produced in more than one pathway with differing number of reaction steps with OH, and those pathways are similar between different aromatic systems.

*Competing interests.* Authors declare no competing interests.

*Author contributions.* AZ, ARK, MB, JEK, KJN, CYL, JCR, JLC, JRR, and MRC collected and analysed data. ARK developed the GKP analysis. FNK and JHK provided project guidance. AZ prepared the manuscript with contributions from all co-authors.

*Acknowledgments.* This work was supported by the Harvard Global Institute, the NSF award AGS-1638672, and a core center
grant P30-ES002109 from the National Institute of Environmental Health Sciences, National Institutes of Health. ARK acknowledges support from the Dreyfus Postdoctoral Program. MB acknowledges support from the Austrian science fund (FWF), grant J-3900.



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





**Table 1: Major toluene oxidation products measured by PTR-MS and NH$_4^+$ CIMS.**

| Product Name | Product Formula | Generation parameter $m$ | Kinetic parameter $k$ (cm$^3$ molecule$^{-1}$ s$^{-1}$) | Yield (%) |
|---|---|---|---|---|
| glyoxal | C$_2$H$_2$O$_2$ | 1.2 | 1.3·10$^{-11}$ | 28% |
| methylglyoxal | C$_3$H$_4$O$_2$ | 1.3 | 1.1·10$^{-11}$ | 15% |
| butenedial | C$_4$H$_4$O$_2$ | 0.8 | 2·10$^{-11}$ | 8% |
| methylbutenedial | C$_5$H$_6$O$_2$ | 0.7 | 2·10$^{-11}$ | 37% |
| epoxybutanedial | C$_4$H$_4$O$_3$ | 1.3 | 1·10$^{-11}$ | 5% |
| methylepoxybutanedial | C$_5$H$_6$O$_3$ | 1.3 | 1·10$^{-11}$ | 5% |
| benzaldehyde | C$_7$H$_6$O | 1 | 1·10$^{-11}$ | 9% |
| cresol | C$_7$H$_8$O | 1 | 2.3·10$^{-11}$ | 10% |
| dicarbonylepoxide | C$_7$H$_8$O$_3$ | 0.9 | 2.2·10$^{-11}$ | 1% |

**Table 2: Major 124-TMB oxidation products measured by PTR-MS and NH$_4^+$ CIMS**

| Product Name | Product Formula | Generation parameter $m$ | Kinetic parameter $k$ (cm$^3$ molecule$^{-1}$ s$^{-1}$) | Yield (%) |
|---|---|---|---|---|
| methylglyoxal | C$_3$H$_4$O$_2$ | 1.3 | 1.8·10$^{-11}$ | 25% |
| biacetyl | C$_4$H$_6$O$_2$ | 1.3 | 2.2·10$^{-11}$ | 10% |
| methylbutenedial | C$_5$H$_6$O$_2$ | 0.7 | 3.5·10$^{-11}$ | 5% |
| dimethylbutenedial | C$_6$H$_8$O$_2$ | 0.9 | 4.1·10$^{-11}$ | 40% |
| trimethylbutenedial | C$_7$H$_{10}$O$_2$ | 1 | 5.1·10$^{-11}$ | 2% |
| methylepoxybutanedial | C$_5$H$_6$O$_3$ | 1.8 | 1.9·10$^{-11}$ | N/A |
| dimethylepoxybutanedial | C$_6$H$_8$O$_3$ | 1.1 | 2.4·10$^{-11}$ | 2% |
| dimethylbenzaldehyde | C$_9$H$_{10}$O | 1 | 1.4·10$^{-11}$ | 3% |
| trimethylphenol | C$_9$H$_{12}$O | 1 | 3.5·10$^{-11}$ | 2% |
| dicarbonylepoxide | C$_9$H$_{12}$O$_3$ | 1 | 4·10$^{-11}$ | 1% |



**Table 3: Kinetics fit of gas-phase highly oxygenated compounds detected in toluene (top five rows) and 1,2,4-TMB (bottom seven rows) oxidation experiments.**

| Product Formula | Generation parameter $m$ | Kinetic parameter $k$ (cm$^3$ molecule$^{-1}$ s$^{-1}$) |
|:---:|:---:|:---:|
| $C_7H_8O_4$ | 1.5 | $1 \cdot 10^{-11}$ |
| $C_7H_{10}O_5$ | 1.5 | $1 \cdot 10^{-11}$ |
| $C_7H_8O_6$ | 1.6 | $1.1 \cdot 10^{-11}$ |
| $C_7H_9NO_6$ | 2.1 | $0.8 \cdot 10^{-11}$ |
| $C_7H_9NO_8$ | 2 | $0.9 \cdot 10^{-11}$ |
| $C_9H_{12}O_4$ | 1.5 | $1.8 \cdot 10^{-11}$ |
| $C_9H_{14}O_4$ | 1.6 | $1.7 \cdot 10^{-11}$ |
| $C_9H_{14}O_5$ | 1.8 | $1.9 \cdot 10^{-11}$ |
| $C_9H_{12}O_6$ | 1.8 | $1.4 \cdot 10^{-11}$ |
| $C_9H_{13}NO_6$ | 1.4 | $1.7 \cdot 10^{-11}$ |
| $C_9H_{13}NO_7$ | 2.1 | $2.3 \cdot 10^{-11}$ |
| $C_9H_{13}NO_8$ | 2.3 | $1.8 \cdot 10^{-11}$ |


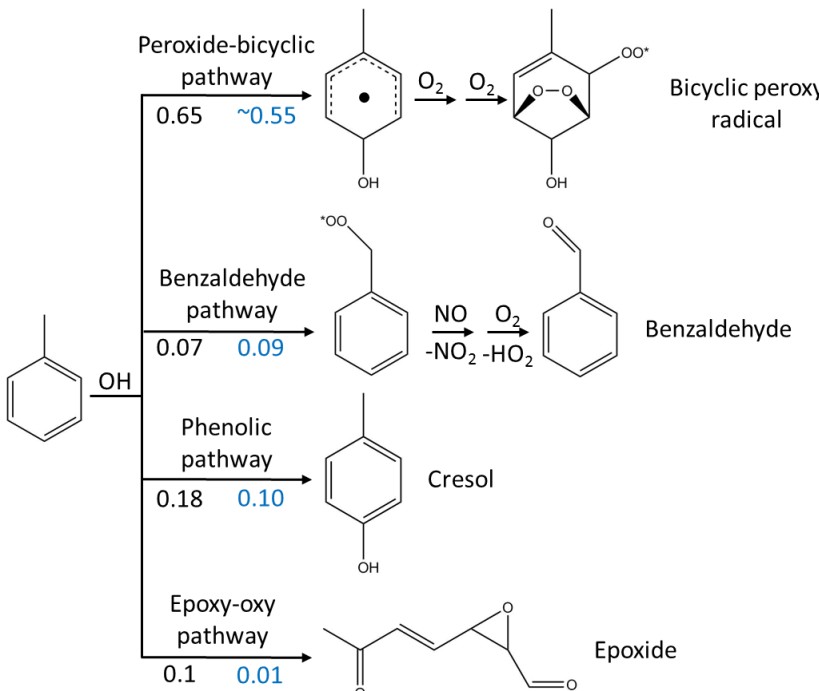

**Figure 1: Gas-phase chemical mechanism for toluene photooxidation. Reaction yields for the major oxidation pathways recommended by MCM v3.3.1 are shown in black. The proposed yields from the present study are shown in blue. The yield of the peroxide-bicyclic pathway is calculated based on the yields of ring-scission products.**



**Figure 2: Oxidation pathways of bicyclic peroxy radicals in the OH-initiated oxidation of 1,2,4-trimethylbenzene. The starting radical us shown in blue.**





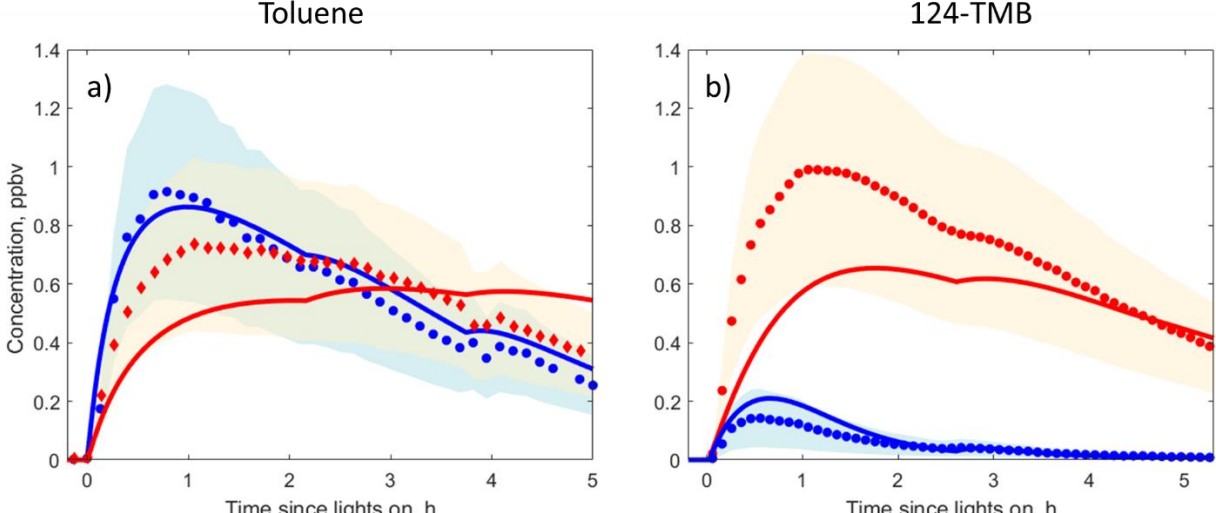

**Figure 3: MCM v3.3.1 predictions (solid lines) compared to PTR-MS (circles) and NH$_4^+$ CIMS (diamonds) measurements under high-NO$_x$ oxidation of (a) toluene and (b) 124-TMB for phenols (red) and benzaldehydes (blue). The uncertainty in the PTR-MS and NH$_4^+$ CIMS measurements is shown in blue and red shading.**



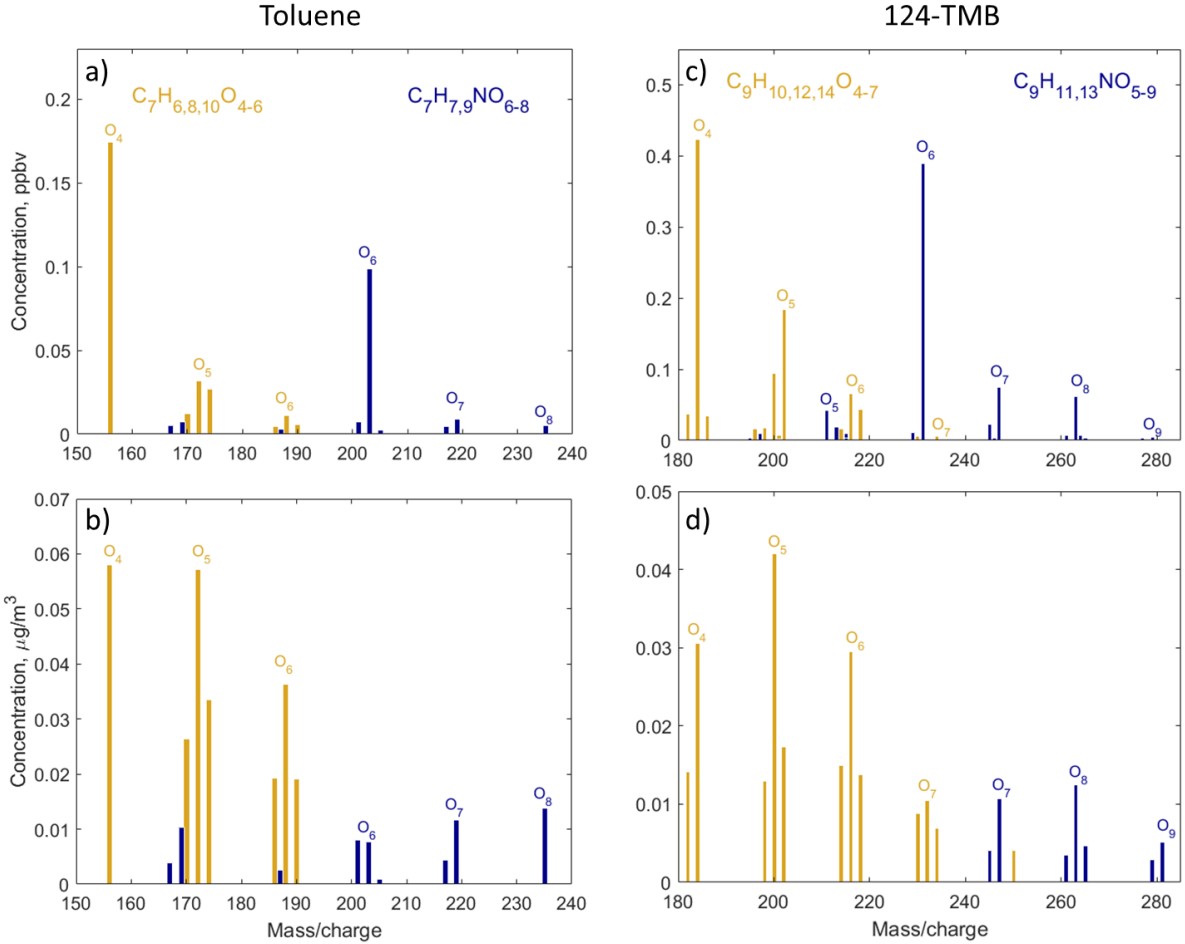

**Figure 4: Mass spectra of compounds with high oxygen content (O:C > 0.44) from toluene (panels a and b) and 1,2,4-trimethylbenzene (panels c and d) detected by NH$_4^+$ CIMS. Gas-phase mass spectra are shown in the upper panels (a) and (c), and particle-phase mass spectra are shown in bottom panels (b) and (d). Non-nitrogen-containing compounds are shown in yellow, and nitrogen-containing compounds are shown in blue.**


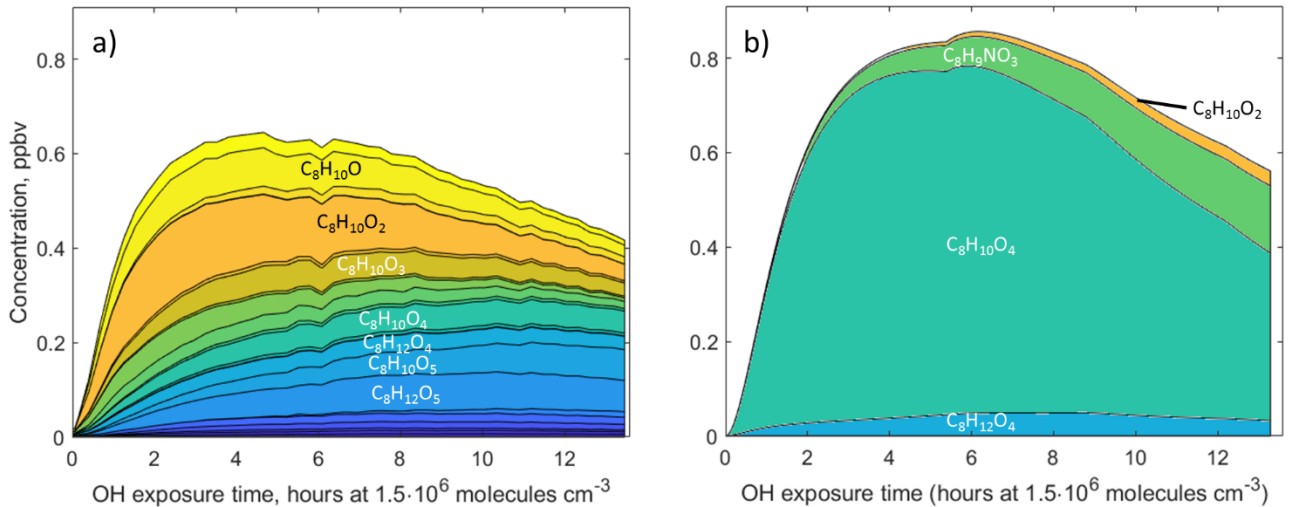

**Figure 5: C₈ gas-phase products (a) detected by PTR-MS and NH₄⁺ CIMS and (b) predicted by MCM v3.3.1 during oxidation of 1,2,4-trimethylbenzene.**




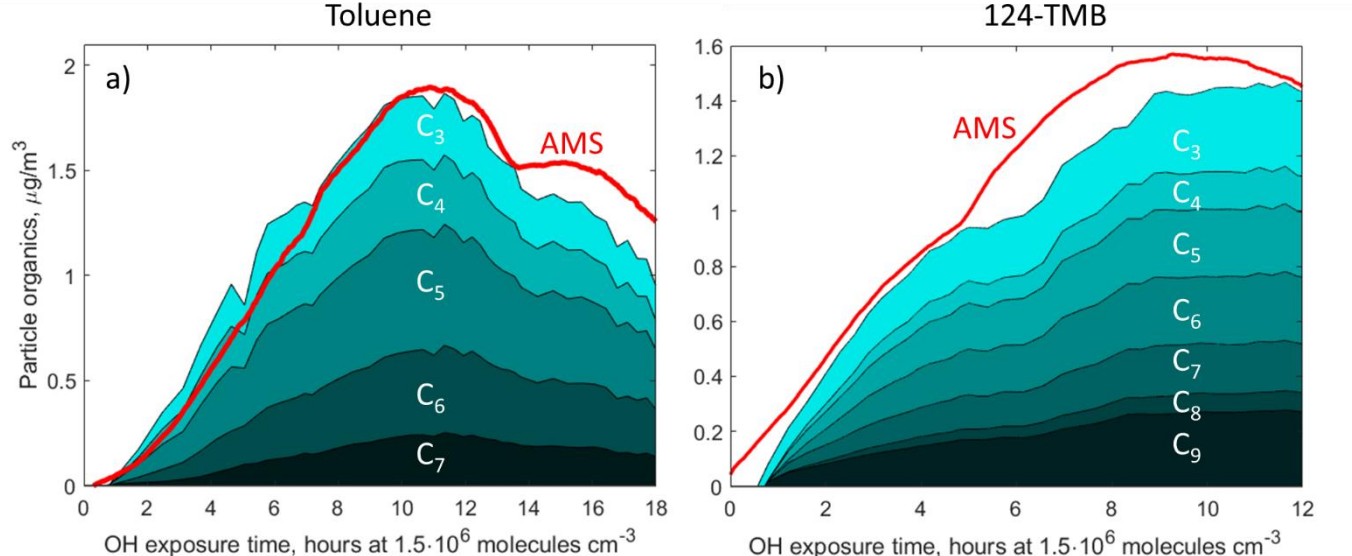

**Figure 6: Total organics measured in the particle phase by NH₄⁺ CIMS and binned by the carbon atom number (a) in the 124-TMB photooxidation experiment and (b) the toluene photooxidation experiment. Total carbon measured by AMS is in red.**