# Peer review of "Mechanistic Study of Formation of Ring-retaining and Ring-opening Products from Oxidation of Aromatic Compounds under Urban Atmospheric Conditions"

_Atmospheric Chemistry and Physics, 2019_

## Referee Comment (RC1) · Anonymous Referee #1 · 20 Aug 2019

Zaystev et al. report interesting experimental and mechanistic studies of products originated from the oxidation of toluene and 1,2,4-trimethylbenzene with OH radicals. The authors make use of several instruments based on chemical ionization coupled to high resolution mass spectrometry (CIMS and PTR). The experimental work is based on conducting photooxidation experiments, followed by the analysis of gas and particle phase products. The analysis is based on chemical ionization techniques in the gas phase using several reagent agents to target specific class of compounds. A complementary mechanistic work (kinetic model and Gamma kinetics parametrization) was

used to model the experimental data. Although these experimental techniques provide powerful information about the formulas, they lack definitive structural information. I do have several concerns associated with the present manuscript:

1. Clarity of the manuscript, the novelty, and the technical interpretation of the data. The authors need to do more through job to clearly describe the objective of the study. Why gas phase and particle phase were measured? In several occasion, I got lost and it was difficult to follow which instrument/reagent agent was used for what and why (a table in the SI will be beneficial for example)? The gas phase was conducted to help with mechanism, however the particle phase was also analyzed but was not discussed (why and how these particle species were formed: partitioning, heterogeneous chemistry...). Organonitrates were mentioned but not discussed! The uncertainties were not discussed here (see my minor comments). Toluene and 124-TMS were studied in the literature and several important studies were not reported/discussed (e.g. Kamens/Jang group on toluene, Kleindienst work, Laskin work etc..). Several papers report chemical species observed in toluene SOA and in ambient PM2.5 in several places around the world. Do these species were observed here? 2. Another important point that need to be discussed by the authors: artefacts associated with these techniques mainly thermal degradations of polar compounds/side chemical reactions (see recent papers by Jimenez and P. Ziemann groups for example). I have provided two references below (some co-authors in this study were also associated with these artefacts studies: a. Xiaoxi Liu, Benjamin Deming, Demetrios Pagonis, Douglas A. Day, Brett B. Palm, Ranajit Talukdar, James M. Roberts, Patrick R. Veres, Jordan E. Krechmer, Joel A. Thornton, Joost A. de Gouw, Paul J. Ziemann, and Jose L. Jimenez b. Effects of gas–wall partitioning in Teflon tubing and instrumentation on time-resolved measurements of gas-phase organic compounds. Demetrios Pagonis, Jordan E. Krechmer, Joost de Gouw, Jose L. Jimenez, and Paul J. Ziemann

3. Given the analysis of gas and particle phase species in this work, it would be useful to include table(s) that outline partitioning coefficients of these species. A discussion

will be beneficial here since gas and particle are linked together! How this study could be beneficial to urban atmosphere and the contribution of aromatic to ambient organic aerosol? Are these HOMs species important? While this study might provide valuable information for a better understanding of the chemical pathways from the photooxidation of toluene and 124-TMB, the results presented here are not sufficiently discussed and/or do not present a real novelty (most products observed here were reported in the literature!).

other comments. 1. Abstract 1. Page 1, line 14. Change "A series of" with "eight" report how many experiment were conducted. 2. Page 1, line 17. Need to add the reagent chemicals used for the two PTRs to be consistent with the CIMS (e.g. H3O+ PTR, . . .). 3. The end of the following sentence is hard to understand "An extensive suite of instrumentation including two Proton-Transfer Reaction Mass-Spectrometers (PTR-MS) and two Chemical Ionization Mass-Spectrometers (NH4+ CIMS and I- CIMS) allowed for quantification of reactive carbon in multiple generations of oxidation"

2. Introduction 4. Page 2, line 5. Needs reference(s) 5. Page 2, line 9-10. The authors state that 124-TMB is a good candidate (serve a model molecule) for substituted aromatic compounds? Please add reference(s) here and clarify this statement? This class of compounds behave widely differently vis-à-vis "chemistry, SOA production, OH rate constants. . ." 6. Page 2, line 11. Delete "often" 7. Figure 1. Suggest adding a second panel on the right side associated with 124-TMB (similar to the toluene). 8. Figure 2. Is the chemistry under low NOx relevant here (either under the chamber conditions or under urban high NOx conditions)? Hydroperoxide channels are minor! The paper focusses on high NOx! It is unlikely that these chemicals are formed under the conditions presented in this study. I suggest this figure should describes the chemistry relevant to the conditions reported in this study. There is a large number of such mechanisms reported in the literature? 9. Page 2, lines 20-21. Delete the sentence associated with low NOx. 10. Please add "BPRs" to figures. 11. Please correct the title of Figure 2. Also in figures 1 and 2, references should be provided in the titles

since the mechanism presented is not new to this study. Are the structures provided in Figure 2 experimentally determined in the literature? This should be stated clearly if these structures (mainly HOMs) were only proposed based on chemical formulas obtained from HR-CIMS/PTR and mechanism/computational chemistry. 12. Page 3, line 1. Delete "the " in ". . .the detailed mechanism. . ." Please provide experimental evidence that NOx level was 10 ppb. 13. Page 3, line 2. The chamber conditions were stated in several part of the text as high-NOx conditions. Here they introduce "moderate". The description of the conditions should be consistent through out the text. It is confusing and arbitrary throughout the literature how high-NOx and low-NOx conditions are defined? 14. Page 3, lines 1-7. This section does not provide clearly the work conducted in this study. Either the experimental or the mechanistic work? Define that gas phase and particle phase were the focus. In some area of the paper it seems that only the gas phase products were measured and in some other parts the particle phase (SOA) were analyzed. Please clearly define that both gas phase and particle were analyzed using these two instruments running under different reagent chemicals to analyze a wide range of compounds. The CIMS was used to analyze particle phase.

3. Methods 15. Page 3, line 11, please provide the flow rate used to keep the chamber volume constant (this important for dilution)? 16. Are the temperature and RH being constant throughout the experiment? When turning the light ON in general T and RH change due to reactions and heat from the UV lamps? Comment here. Time series of T and RH should be provided in Figure S1 top. 17. Page 3, line 13-14. "We performed a series of photochemical experiments, in which toluene and 1,2,4-TMB were oxidized by OH under high-NO conditions (Table S1)." Please provide NOx (NO and NO2) concentrations in Table S1. What the authors refers to high NOx conditions? Provide references here! 18. Please provide range of RH and T in Table S1 since term "approximative" was used. 19. Page 3, line 14. "First, dry ammonium sulfate particles, used as condensation nuclei, were injected in the chamber reach a number concentration of (2.5 - 5.7)·104 cm-3." Table S1, shows particle loading after seed injection in the order of (2.6 - 5.7).104 cm3 (no changes of the particle number after

the light was ON and within the experimental error). Is this number stays the same throughout the experiment? Please clarify this in the footnote. This is important since HOMs were observed and will indicate if nucleation occurred or just growth of seed particles!!

I'm curious how much aerosol mass was formed? - Table S1 should provide the amount of aerosol formed minus the seed aerosol injected. Specify the time of the reaction for these values? It is not stated in the manuscript that these values are associated with peak aerosol and compounds? - How much toluene was reacted? Since these experiments were conducted at low HC and NOx concentrations, uncertainties should be discussed, and error bars should be provided for toluene and 124TMB. Providing time series in figure S1 top for example or in a separate figure will make the manuscript stronger? 20. Page 1, line 16-17. "Nitrous acid (HONO) was later injected as an OH precursor." It is not clear here! Table S1 shows that HONO was injected initially and during the experiment. Please clarify this statement? Please provide the source of NO in this section? 21. Page 3, line 18-19. Data in Table S1 not consistent with the statement here? HONO in Table S1 was between 28 and 60 ppbv. Either Table S1 or the text needs to be corrected? 22. HONO was injected before the light was on. There is no background OH before the light was ON? Was the chamber in the dark before t=0? Figure S1 bottom shows background for NO2, O3, and HONO+NOX? It is beneficial if C6F6 was presented also in Figure S1 top (dilution). - O3 was high ($\sim$15 ppb) before light was ON! From where O3 was generated before light ON? 23. "The concentration of NO in the chamber was estimated to be $\sim$0.3 ppbv while NO2 concentration was approximately 10 ppbv" I believe this is the initial concentration? In the manuscript the authors describe the experiments were conducted under high-NO? This is confusing? From this statement and Figure S1, the experiments were conducted under high NO2? Please clarify this? How NO was estimated? 24. I suggest adding experiment ID in Table S1. Then, associate the experiment ID in each figure/table and where the data is discussed in the text? It is hard to link the origin of the data when discussed in the text figures and tables? 25. I'm confused with the term initial in Table S1. Are the

authors referring before the light was ON? The system seems to be dynamic (clean-air was continuously injected: dilution always occurring)" The initial concentrations of toluene and 124TMB is very difficult to measure since the system was dynamic? I believe any quantitative data will be associated with high uncertainty? Provide error discussion in the manuscript! When conducting chamber experiments at low HC and NOx concentrations, initial HCs and NOx should be presented prior to light is ON and during the experiment. The instruments used should provide these data? Dilution data (C6F6) also should be provided in Figure S1? 26. Page 3, line 22-23. "The reagents were allowed to mix for several min, . . ." It is not clear if the chamber was mixed well (since no fans were used) before the start of the reaction (light ON?) It is not easy to conduct these kings of experiments: the chamber is always under dilution "continuously with clean air" at the same time injecting a known amount of reagents before the light is ON. It is necessary to provide the time series of all parameters measured and used to conduct the experiment? T, RH, HONO, HCs, SOA formed, C6F6. . . [Note, the amount of HC injected and the volume of the chamber does not provide accurate concentrations!] In general, HCs and HONO as well other reagents should be injected continuously in the chamber until a stable concentration is attained for all reagents, then stop the HC injection and turn ON the light (start of the reaction). Comments from the authors is need here! I do see the use of C6F6 to measure the dilution rate and it was discussed rarely in this paper. 27. Table S1. Particle loading should be (2.6 - 5.7) 104 instead of (2.6 - 3.5)104 as stated in line 15, page 1. 28. Suggest using "Experimental section" instead of "Experimental design" 29. Pages 3-4: chamber instrumentation. The description of the instruments is difficult to follow and switching between CIMS and PTR notations is hard to follow. All instruments use chemical ionization. At the beginning the authors state that four instruments were used (two CIMS and two PTR) and in this section it seems that three instruments were used: (1) page 4, line 1 ("including the I- CIMS instrument; and two PTR (CIMS H3O+ and NH4+. This section needs to be clarified and consistent with the manuscript. Recently artefacts and sampling issues were reported, and this should be discussed in the paper (see

my general comments). 30. What reagent gas was used for the Vocus-2R-PTR? 31. The authors should provide the 10 compounds used in the calibration of I-CIMS and PTR3 in SI. Using these instruments for aerosol characterization may be associated with sampling artefacts as discussed recently (see my general comments) 32. Page 5, lines 4-24. Please provide time series for [ArVOC]0 for toluene and 124-TMB as I suggested before? Data obtained from PTR! 33. Page 5. The yields were described but were not provided in the paper! 34. What "b" represents in eq. 3? 35. Page 4, line 31. Is the light intensity was measured? 36. Page 4, 5: Gamma Kinetic parameterization. Can the structure affect the results (the parameters obtained)? This model is applied to formulas here? 37. Page 6, line 18. Figure 1 should incorporate both HCs as I stated before. 38. Page 10, line 18. Define TD-NH4+ CIMS. 39. Figure 6. Figure 6 title. I think there is an error! (a) should be toluene and (b) should be 124-TMB. Please clarify? Also is total organics or total organics carbon? 40. Page 10. "The O:C ratios calculated from individual species measured from thermally desorbed SOA using NH4+ CIMS were ~0.95 for toluene SOA and ~0.7 for 1,2,4-TMB SOA. These ratios are in good agreement with the atomic O:C ratios measured by AMS (0.85 and 0.65 for toluene and 1,2,4-TMB SOA, respectively) (Canagaratna et al., 2015)." Does this comparison was applied to the same SOA size for the AMS and CIMS? 41. Page 10. "Products observed in the gas phase are compared to those detected in the particle phase to further understand the mechanism of SOA formation from aromatics precursors." Can partitioning coefficients be obtained for these products? Please provide organic species present in both gas phase and particle phase and their estimated partitioning coefficient? Table for example. 42. The section describing on page 10, lines 25 - 32. "non-fragmentary" "ring-retaining" "ring scission" "fragmentary" is not clear to me. Could the authors describe how the thermal fragmentation either in the gas phase or the particle phase reported on page 10 can affect the data (formula and structures) provided in this manuscript? Are the HOMs detected not originated from artefacts in the inlets? These compounds with high O:C ratio are expected to be in the particle phase but numerous studies including this one report them in the gas phase? 43. In

general, chamber backgrounds NOx always is present (although clean air was used) due to wall chemistry and the history of the chamber (heterogeneous wall chemistry). Comments!

---

## Referee Comment (RC2) · Anonymous Referee #2 · 4 Sep 2019

In this work, the authors presented results from oxidation experiments of aromatic compounds, toluene and 1,2,4-TMB. These aromatic compounds are important VOCs in urban areas, and their oxidation leads to significant ozone and secondary organic aerosol (SOA) formation. In this study the authors employed a number of new analytical techniques to measure the gas and particle phase composition, and compared to the latest version of Master Chemical Mechanism (MCM), which summarizes the current understanding about the mechanisms. Furthermore, the time trend analysis using gamma kinetic parameterization is a novel method to look at the multigenerational chemistry.

This manuscript is well written, and I only have some minor suggestions. I recommend publication of this manuscript in ACP.

Minor scientific comments:

1. It would be good to know on a bulk or general level, how these results improve the understanding of the chemistry. For example, I wonder what the carbon closure now is, with these new measurements. Figure 3 is probably a good place to show that.

2. Somewhat related: One key piece of information shown in Section 3.3 and Fig. 6 is that the total SOA mass measured by AMS and NH4 CIMS compare very well, and so do the O/C ratios. This is an important discovery and should be highlighted in the abstract.

3. The multigenerational chemistry of many of the products is a key contribution. I expect that the accompanying paper describing the methods will be well received. There are some ambiguous ones that have non-integer m (e.g. 1.7-1.8). What is the general uncertainty in this analysis?

4. Related to comment/question 3: I expect that some experiments with oxidation of later generation products would be very helpful. For example, oxidation of cresol (which is commercially available) should yield lower m for some of the products. Perhaps even examining the decrease in m would help apportion the relative amount for each generation. I think these are important experiments anyway given that the authors are claiming the importance of phenolic and benzaldehyde pathways in HOM production.

5. The experiments were all conducted under RH of 2%. While I completely understand the rationale to create a well-controlled environment, it may be worthwhile to mention this is a potential limitation of this study and discuss implications. I do not see water playing an important role in the gas-phase chemistry, but could potentially shorten the lifetime of particle-phase hydroperoxides, epoxides and organic nitrates.

Minor technical comments:

Methods: I do not understand why the authors would use hexafluorobenzene as a tracer for both chamber wall loss and dilution of VOCs. I can see hexafluorobenzene is a good tracer for dilution, but I do not expect it to be lost to the chamber walls. Based on the chamber volume and air refilling rate, the dilution rate can be estimated. Is the hexafluorobenzene decaying faster than this dilution rate? If so, why is it being lost to the walls?

CO and formaldehyde were mentioned in methods, but no results were presented.

Methods: Particle-phase compounds were quantified using I- CIMS, but for the gas phase compounds the authors claim I- CIMS is quite uncertain. Are the uncertainties in quantification the same for both phases?

Section 2.4: how large are the time steps?

Section 3.1: Is it possible that the epoxide was not detected because of thermal de-composition for the particle phase measurements, or fragmentation during ionization?

Section 3.2.1, Line 13: BPR has been defined earlier.

Section 3.2.1 Line 29-30: Presumably the lifetimes are calculated using generic RO2+NO and RO2+HO2 rate constants? What rate constants were used?

Tables 1-3: what are the uncertainties in m from the fits?

Figure 6: why is there a discontinuity at 14 hours of exposure for toluene? Was more OH precursor added? Similarly there seems to be one as well for TMB at 4 h.

---

## Author Comment (AC1) · 5 Nov 2019

**Response to Reviewer 1**

Reviewer comments are in **bold**. Author responses are in plain text. Excerpts from the manuscript are in *italics*. Modifications to the manuscript are in *blue italics*. Page and line numbers in the responses correspond to those in the original ACPD paper.

**Zaystev et al. report interesting experimental and mechanistic studies of products originated from the oxidation of toluene and 1,2,4-trimethylbenzene with OH radicals. The authors make use of several instruments based on chemical ionization coupled to high resolution mass spectrometry (CIMS and PTR). The experimental work is based on conducting photooxidation experiments, followed by the analysis of gas and particle phase products. The analysis is based on chemical ionization techniques in the gas phase using several reagent agents to target specific class of compounds. A complementary mechanistic work (kinetic model and Gamma kinetics parametrization) was used to model the experimental data. Although these experimental techniques provide powerful information about the formulas, they lack definitive structural information. I do have several concerns associated with the present manuscript:**

We thank the reviewer for the input that helped us to improve the manuscript. The revised manuscript takes into account the comments and questions, as detailed in the responses below.

We break up comment 1 into individual sub-comments and respond to them separately.

**1a. Clarity of the manuscript, the novelty, and the technical interpretation of the data. The authors need to do more through job to clearly describe the objective of the study. Why gas phase and particle phase were measured?**

The objective of this study is to evaluate the importance of various gas-phase oxidation pathways of aromatic compounds in terms of production of oxygenated low-volatile species (including HOMs) and SOA formation potential. In response to the comment, we update the following sentence (P3 L5):

*The goal of this work is to  identify gas-phase pathways leading to production of low-volatility compounds which are important for SOA formation and support these identifications with CIMS data and a method to characterize the kinetics of an oxidation system.*

**1b. In several occasion, I got lost and it was difficult to follow which instrument/reagent agent was used for what and why (a table in the SI will be beneficial for example)?**

In a typical 1,2,4-TMB experiment 570 ions detected by $NH_4^+$ CIMS and 590 ions detected by $H_3O^+$ CIMS were enhanced during the experiment. Hence, we think that it would be overwhelming to include them as a table in SI. However, in response to the comment, we clarify which instrument and reagent ion were used throughout the manuscript:

P6 L24: *In the toluene experiments, the approximate yields of benzaldehyde and cresol (~0.10 and ~0.16, respectively) were calculated based on the decay of toluene measured by Vocus-2R-PTR-TOF, rise of the two products measured by PTR3 $H_3O^+$ CIMS and $NH_4^+$ CIMS (cresol was measured by $H_3O^+$ CIMS while $NH_4^+$ CIMS was used for detecting benzaldehyde), and accounting for losses of cresol and benzaldehyde from wall deposition and reaction with OH and $NO_3$ (Sect 2.4).*

P7 L2: *The kinetic model predictions for the two products agree within uncertainties with the  $H_3O^+$*

*CIMS measurements and the time series behaviour is again similar (Fig. 3b).*

P7 L8: *The  yields of these products observed with NH₄⁺ CIMS are significantly smaller (~0.01 in both systems).*

P8 L33: *The total concentration of C₈ components predicted by MCM v3.3.1 for 1,2,4-TMB is in good agreement with the NH₄⁺ CIMS and  H₃O⁺ CIMS measurements (less oxidized compounds (O:C < 0.25) were detected using H₃O⁺ CIMS while NH₄⁺ CIMS was used for more oxidized species), though the observed composition is distinctly different from the MCM prediction (Fig. 5).*

**1c. The gas phase was conducted to help with mechanism, however the particle phase was also analyzed but was not discussed (why and how these particle species were formed: partitioning, heterogeneous chemistry…).**

The goal of this study is to evaluate the importance of various gas-phase oxidation pathways of aromatic compounds in terms of SOA formation potential. In Section 3.3 "SOA Analysis" products detected in the particle phase are compared to those detected in the gas phase to further understand the mechanism of toluene and 1,2,4-TMB SOA formation (P10 L24).

**1d. Organonitrates were mentioned but not discussed!**

While we agree with the reviewer that organonitrates make up an important class of organic molecules in the atmosphere, discussing individual classes of atmospheric compounds lies beyond the scope of this manuscript. The goal of this study is rather to evaluate the importance of various gas-phase oxidation pathways leading to the formation of highly oxygenated, low-volatile compounds that might contribute to SOA formation.

**1e. The uncertainties were not discussed here (see my minor comments).**

Uncertainties of the used instrumentation are discussed in Section 2.2 (P4 L14-16).

**1f. Toluene and 124-TMS were studied in the literature and several important studies were not reported/discussed (e.g. Kamens/Jang group on toluene, Kleindienst work, Laskin work etc..). Several papers report chemical species observed in toluene SOA and in ambient PM2.5 in several places around the world. Do these species were observed here?**

We think it would be overwhelming to include all existing references on photooxidation of toluene, 124-TMB and other aromatic compounds in this short paper. We included what we thought were the most relevant references for understanding the significance of various oxidation pathways of aromatic hydrocarbons. In addition, Calvert et al. (2002) cited in the manuscript gives an excellent overview of the relevant literature prior to 2002. However, in response to this comment, we added a few additional references, including previous toluene photooxidation studies (P2 L14):

*The OH-adducts can react with atmospheric O₂ through H-abstraction to form ring-retaining phenolic compounds (i.e., cresols and trimethylphenols) (Jang and Kamens, 2001; Kleindienst et al., 2004).*

**2. Another important point that need to be discussed by the authors: artefacts associated with these techniques mainly thermal degradations of polar compounds/side chemical reactions (see recent papers by Jimenez and P. Ziemann groups for example). I have provided two references below (some co-authors in this study were also associated with these artefacts studies: a. Xiaoxi Liu, Benjamin Deming, Demetrios Pagonis, Douglas A. Day, Brett B. Palm, Ranajit Talukdar, James M. Roberts, Patrick**

**R. Veres, Jordan E. Krechmer, Joel A. Thornton, Joost A. de Gouw, Paul J. Ziemann, and Jose L. Jimenez**
**b. Effects of gas–wall partitioning in Teflon tubing and instrumentation on time-resolved measurements of gas-phase organic compounds. Demetrios Pagonis, Jordan E. Krechmer, Joost de Gouw, Jose L. Jimenez, and Paul J. Ziemann**

In response to the comment, we discuss three types of potential artefacts associated with CIMS technique: thermal degradation, fragmentation during ionization, and inlet losses.

a. During particle-phase measurements using TD-NH$_4^+$ CIMS and TD-H$_3$O$^+$ CIMS, sampled air passes through a gas-phase denuder (Ionicon Analytik GmbH, Austria), which removes the gas-phase organics, and then through a thermal desorption region heated to 180℃, in which the aerosol particles are vaporized. This may result in thermal decomposition of OVOCs. We studied thermal decomposition of OVOCs extracted from alpha-pinene SOA by measuring their peak intensities using TD-NH$_4^+$ CIMS. Signals of many species increased at moderate temperatures ($T$ < 160°C) and levelled out or decreased at higher temperatures ($T$ > 180°C), as shown in Fig. R1 (Zaytsev et al., 2019). Therefore, we chose TDU temperature to be 180℃, as at this temperature the major fraction of particles was evaporated while thermal decomposition of labile species was relatively small.

[Figure]

Figure R1: Thermograms of select OVOCs extracted from alpha-pinene ozonolysis SOA.

We modify the description of TD-NH$_4^+$ CIMS and TD-H$_3$O$^+$ CIMS (P4 L21):

*Particle-phase compounds were quantified using the FIGAERO-HRToF-I$^-$ CIMS (Lopez-Hilfiker et al., 2014), and a second PTR3 that could be operated in two positive modes as described above and equipped with an aerosol inlet comprising a gas-phase denuder and a thermal desorption unit heated to 180℃ (TD-NH$_4^+$ CIMS and TD-H$_3$O$^+$ CIMS). At this temperature, all particles were evaporated while thermal decomposition of labile oxygenated compounds was relatively small (Zaytsev et al., 2019).*

b. Electric field strength within the CIMS instruments controls the amount of gas phase ion clustering and fragmentation which may lead to additional artefacts in the gas-phase data. Moderately strong fields are routinely used in H$_3$O$^+$ CIMS ($E/N$ = 90 Td in PTR3 H$_3$O$^+$ CIMS) which may result in fragmentation of analyte molecules. In particular, large hydrocarbons including aromatic

compounds ($C_8$ and larger) are known to fragment to small common masses, while aldehydes and alcohols can lose $H_2O$ (Erickson et al., 2014; Buhr et al., 2002). On the contrary, weaker fields are used in $NH_4^+$ CIMS ($E/N$ = 60 Td in PTR3 $NH_4^+$ CIMS) and therefore result in smaller fragmentation comparing to $H_3O^+$ CIMS. Hence, PTR3 $NH_4^+$ CIMS was mainly used for detection of larger and more functionalized molecules (P4 L11).

c. The following discussion of inlet losses is included in the Supplement:

*Inlet losses in CIMS instruments*
*Concentrations of gaseous analytes can be perturbed by gas-wall interactions occurring in the tubing used for sampling gases from the environmental chamber or inside the instruments. In order to estimate the response timescale of CIMS instruments (PTR3, Vocus, and I⁻ CIMS), we follow the procedure described in detail by Pagonis et al. (2017). At the beginning of the procedure, the instruments sampled air from the environmental chamber containing decane photooxidation products. Later, the instruments sampling lines were abruptly reconnected to zero air resulting in step-function decrease in the concentrations of the oxidation products. PTR3, Vocus and I⁻ CIMS time responding profiles measured in response to this step-function decrease were used to calculate delay times for the three instruments. Inlet delay times for all measured compounds did not exceed 20 seconds (Fig. S1).*

[Figure]

*Figure S1: Measured instrument delay times as a function of SIMPOL c\* for CIMS instruments.*

The following sentence is changed (P4 L6):

*PTR3 and Vocus-2R-PTR-TOF are designed to minimize inlet losses of sampled compound (Krechmer et al., 2018; Breitenlechner et al., 2017); for more details see the Supplement.*

**3. Given the analysis of gas and particle phase species in this work, it would be useful to include table(s) that outline partitioning coefficients of these species. A discussion will be beneficial here since gas and particle are linked together! How this study could be beneficial to urban atmosphere and the contribution of aromatic to ambient organic aerosol? Are these HOMs species important? While this study might provide valuable information for a better understanding of the chemical pathways from**

**the photooxidation of toluene and 124-TMB, the results presented here are not sufficiently discussed and/or do not present a real novelty (most products observed here were reported in the literature!).**

Partitioning coefficients for low-volatile compounds could be obtained if the equilibrium between the gas and particle-phase organics was achieved. However, there are a number of reasons why simple gas-particle partitioning might not explain the concentrations observed in the two phases. First, chamber lights were turned on during the whole duration of experiments resulting in constant production and loss of oxidation products in the gas phase. Second, particles were lost on the chamber walls, while the wall loss term was not estimated. Third, CIMS and PTR-MS instruments cannot distinguish between isomers and do not provide further insight into the molecular structure of detected compounds. Hence, it is possible that several compounds with the same molecular formulas, but different structures and partitioning coefficients were lumped together. Therefore, partitioning coefficients for observed species cannot be reported.

While we agree with the reviewer that the majority of the observed products had been reported before, the goal of this study was to identify gas-phase pathways leading to production of low-volatility compounds which are important for SOA formation and support these identifications with CIMS data and a novel method to characterize the kinetics of an oxidation system.

**Other comments.**

**1. Page 1, line 14. Change "A series of" with "eight" report how many experiment were conducted.**

We change the sentence as suggested (P1 L14):

*Four  toluene and four 1,2,4-trimethylbenzene (1,2,4-TMB) photooxidation experiments were performed in an environment chamber under relevant polluted conditions ($NO_x$ ~ 10 ppb).*

**2. Page 1, line 17. Need to add the reagent chemicals used for the two PTRs to be consistent with the CIMS (e.g. $H_3O^+$ PTR, ...).**

Unlike CIMS instruments, which can use a variety of reagent ions to detect different classes of VOCs, PTR-MS instruments routinely use hydronium ($H_3O^+$) ions to ionize VOCs (Yuan et al., 2017). Hence, we think it would be excessive to include reagent ions used by the PTR-MS instruments in the abstract. This information, however, is included in Section 2.2 "Chamber Instrumentation" (P4 L2-4).

**3. The end of the following sentence is hard to understand "An extensive suite of instrumentation including two Proton-Transfer Reaction Mass-Spectrometers (PTR-MS) and two Chemical Ionization Mass-Spectrometers ($NH_4^+$ CIMS and $I^-$ CIMS) allowed for quantification of reactive carbon in multiple generations of oxidation"**

In this study, we define "generation" as the number of reactions with OH (P6 L13). We change the sentence as following (P1 L16):

*An extensive suite of instrumentation including two Proton-Transfer Reaction Mass-Spectrometers (PTR-MS) and two Chemical Ionization Mass-Spectrometers ($NH_4^+$ CIMS and $I^-$ CIMS) allowed for quantification of reactive carbon in multiple generations of hydroxyl radical (OH)-initiated oxidation.*

**4. Page 2, line 5. Needs reference(s)**

We include the following references (P2 L5):

*Toluene, the most abundant alkylbenzene in the atmosphere, is primarily emitted by aforementioned anthropogenic processes (Wu et al., 2014).*

**5. Page 2, line 9-10. The authors state that 124-TMB is a good candidate (serve a model molecule) for substituted aromatic compounds? Please add reference(s) here and clarify this statement? This class of compounds behave widely differently vis-à-vis "chemistry, SOA production, OH rate constants…"**

The most abundant aromatic species in the atmosphere are benzene, toluene, xylenes, ethylbenzene and trimethylbenzenes (Birdsall and Elrod, 2011). Among these species, trimethylbenzenes represent the most substituted aromatic compounds. In this study we compare and contrast gas-phase oxidation pathways leading to production of low-volatility compounds for less and more substituted aromatics (toluene and 1,2,4-TMB; P3 L1-2).

We change the sentence as suggested (P2 L9):

*1,2,4-trimethylbenzene (1,2,4-TMB) is chosen  as a model molecule to study oxidation of more substituted aromatic compounds (i.e., trimethylbenzenes).*

**6. Page 2, line 11. Delete "often"**

We change the sentence (P2 L11):

*In the atmosphere, oxidation of aromatic hydrocarbons is primarily  initiated by their reactions with hydroxyl radicals (OH) via H-abstraction from the alkyl groups or OH addition to the aromatic ring.*

**7. Figure 1. Suggest adding a second panel on the right side associated with 124-TMB (similar to the toluene).**

We include Figure 1 in this paper to illustrate four major oxidation pathways common for numerous aromatic compounds (including toluene and 1,2,4-TMB). Hence, we think it would be overwhelming to add 1,2,4-TMB oxidation scheme to this figure. However, in response to this comment, we change the title of Figure 1:

*Figure 1:  Major gas-phase oxidation pathways for aromatic hydrocarbons, using toluene as an example. Reaction yields for the  oxidation pathways of toluene recommended by MCM v3.3.1 are shown in black. The proposed yields from the present study are shown in blue. The yield of the peroxide-bicyclic pathway is calculated based on the yields of ring-scission products.*

**8. Figure 2. Is the chemistry under low $NO_x$ relevant here (either under the chamber conditions or under urban high NOx conditions)? Hydroperoxide channels are minor! The paper focusses on high $NO_x$! It is unlikely that these chemicals are formed under the conditions presented in this study. I suggest this figure should describes the chemistry relevant to the conditions reported in this study. There is a large number of such mechanisms reported in the literature?**

We agree with the reviewer that the focus of this paper is the mechanistic study of highly oxygenated, low volatile products from oxidation of aromatic compounds under urban conditions. However, MCM v3.3.1 predicts than under studied conditions a non-negligible fraction of peroxy radicals reacts with $HO_2$

to produce hydroperoxides and carbonyls (Figs. S3 and S4). Therefore, it would be beneficial for the paper to keep the hydroperoxide channel in the Introduction and in Fig. 2. In addition, the formation of compounds with molecular formulas corresponding to hydroperoxides was observed in our experiments (P8 L5-6).

**9. Page 2, lines 20-21. Delete the sentence associated with low NO$_x$.**

As we mentioned in our response to the previous comment, MCM v3.3.1 predicts that under studied conditions the RO$_2$+HO$_2$ reaction can make up 5-10% of the bicyclic peroxy radical reactivity (Figs. S3 and S4). Hence, we think it would be beneficial to discuss all oxidation pathways in the Introduction. However, in response to the comment we change the order of the discussion starting with the reaction with NO and continuing with other reactions (P2 L20):

*Under urban-relevant high-NO conditions BPRs also react with NO to form bicyclic oxy radicals that decompose to ring scission carbonylic products such as (methyl) glyoxal and biacetyl. Recent theoretical studies predict a new type of epoxy-dicarbonyl products that have not reported in previous studies (Li and Wang, 2014, Wu et al., 2014). Reaction of BPRs with NO can also result in formation of bicyclic organonitrates. In addition,*  *BPRs react with HO$_2$ and RO$_2$, forming bicyclic hydroperoxides and bicyclic carbonyls, respectively (Fig. 2). Finally,*  *BPRs can undergo unimolecular H-migration followed by O$_2$-addition ( autooxidation) leading to the formation of non-aromatic ring-retaining highly oxygenated organic molecules (HOMs) (Bianchi et al., 2019). Molteni et al. (2018) reported elemental composition of the HOMs from a series of aromatic compounds produced under low-NO conditions. The autooxidation pathway might be more important for the substituted aromatics because of the higher yield of BPR formation and the larger number of relatively weak C-H bonds (Wang et al., 2017).* ~~*Under urban-relevant high-NO conditions BPRs also react with NO to form bicyclic oxy radicals that decompose to ring scission carbonylic products such as (methyl) glyoxal and biacetyl. Recent theoretical studies predict a new type of epoxy-dicarbonyl products that have not reported in previous studies (Li and Wang, 2014, Wu et al., 2014). Reaction of BPRs with NO can also result in formation of bicyclic organonitrates.*~~

**10. Please add "BPRs" to figures.**

We add the notation "BPR" after "Bicylic peroxy radical" on Fig 1. Figure 2 depicts oxidation pathways of bicylic peroxy radicals, as mentioned in the title of the figure.

**11. Please correct the title of Figure 2. Also in figures 1 and 2, references should be provided in the titles since the mechanism presented is not new to this study. Are the structures provided in Figure 2 experimentally determined in the literature? This should be stated clearly if these structures (mainly HOMs) were only proposed based on chemical formulas obtained from HR-CIMS/PTR and mechanism/computational chemistry.**

We include references and update titles of Figs. 1 and 2 as suggested:

*Figure 1:*  *Major gas-phase oxidation pathways for aromatic hydrocarbons using the example of toluene. Reaction yields for the toluene*  *oxidation pathways recommended by MCM v3.3.1 are shown in black (Bloss et al., 2005). The proposed yields from the present study are shown in blue. The yield of the peroxide-bicyclic pathway is calculated based on the yields of ring-scission products.*

*Figure 2: Oxidation pathways of bicyclic peroxy radicals in the OH-initiated oxidation of 1,2,4-trimehtylbenzene. The starting radical is shown in blue. Bimolecular reactions are from MCM v3.3.1 (Bloss et al., 2005) and Birdsall and Elrod (2011).*

**12. Page 3, line 1. Delete "the " in "…the detailed mechanism…" Please provide experimental evidence that $NO_x$ level was 10 ppb.**

We change the sentence as suggested (P3 L1):

*In the present work, we investigate  detailed mechanism**s** of hydroxyl radical multigeneration oxidation chemistry of two aromatic hydrocarbons: toluene and 1,2,4-trimethylbenzene under moderately high , urban-relevant $NO_x$ levels (~10 ppbv).*

The detailed discussion of chamber conditions during experiments is given in Section 2.1 "Experimental design". As stated in Section 2.2, HONO+$NO_x$ levels were measured by 42i NOx monitor (Thermo Fisher Scientific), and the sum of concentrations of HONO and $NO_x$ concentration for a typical experiment is shown in Fig. S2. However, in response to the comment, we update the title of Fig. S2 to clarify that $NO_2$ concentration shown in the figure was estimated using MCM v3.3.1 and not measured directly:

*Figure S2 : (a) OH exposure,  (b) concentrations of $O_3$, $NO_2$, HONO+$NO_x$, and (c) temperature and RH for a typical photooxidation experiment. $NO_2$ concentration was estimated using F0AM (Wolfe et al., 2016) based on MCM v3.3.1 (Bloss et al., 2005).*

**13. Page 3, line 2. The chamber conditions were stated in several part of the text as high-$NO_x$ conditions. Here they introduce "moderate". The description of the conditions should be consistent through out the text. It is confusing and arbitrary throughout the literature how high-$NO_x$ and low-$NO_x$ conditions are defined?**

We agree with the reviewer that the terms "high-$NO_x$" and "low-$NO_x$" can be arbitrary, so we describe the chamber conditions as "high-NO" throughout the manuscript (e.g., P3 L13). In this work, we follow the definition of high-NO and low-NO chemistry regimes introduced by Seinfeld and Pandis (2016). At sufficiently high-NO levels, the primary chain terminating reaction for peroxy radicals is the $RO_2$ + NO. This condition is called high-NO. On the contrary, if the fate of peroxy radicals is not determined by their reaction with NO, the condition is called low-NO. Under our experimental conditions, the $RO_2$ + NO reaction is estimated to be the primary loss of bicyclic peroxy radicals (Figs. S3 and S4). Hence, we call these experiments high-NO. At the same time, $NO_2$ levels observed in the environmental chamber are relevant for urban conditions as $NO_2$ concentration in major European cities varies from 3-38 ppb (Hazenkamp-von Arx et al., 2004).

We clarify the definition of the high-NO regime in the manuscript (P3 L13):

*We performed a series of photochemical experiments, in which toluene and 1,2,4-TMB were oxidized by OH  (Table S1). The experiments were carried out under high-NO conditions such that the fate of peroxy radicals was primarily determined by their reaction with NO (Seinfeld and Pandis, 2016).*

We change the following sentence (P3 L1):

*In the present work, we investigate  detailed mechanism*s *of hydroxyl radical multigeneration oxidation chemistry of two aromatic hydrocarbons: toluene and 1,2,4-trimethylbenzene under moderately high,  urban-relevant NO*$_x$ *levels (~10 ppbv).*

**14. Page 3, lines 1-7. This section does not provide clearly the work conducted in this study. Either the experimental or the mechanistic work? Define that gas phase and particle phase were the focus. In some area of the paper it seems that only the gas phase products were measured and in some other parts the particle phase (SOA) were analyzed. Please clearly define that both gas phase and particle were analyzed using these two instruments running under different reagent chemicals to analyze a wide range of compounds. The CIMS was used to analyze particle phase.**

In this work we focus on studying both gas- and particle-phase oxidation products: gas-phase products are discussed in Sections 3.1 and 3.2 while particle-products are discussed in Section 3.3.

In response to the comment, we edit the sentence describing the instrumentation used in this study and products studied here (P3 L4):

* We use three chemical ionization mass spectrometry (CIMS) techniques (I*$^-$ *reagent ion, NH*$_4^+$ *reagent ion, and H*$_3$*O*$^+$ *reagent ion) to characterize and quantify gas-phase oxidation products. In addition, NH*$_4^+$ *CIMS and I*$^-$ *CIMS were used to detect particle-phase products.*

**15. Page 3, line 11, please provide the flow rate used to keep the chamber volume constant (this important for dilution)?**

We include the clean air flow rate in the manuscript (P3 L11):

*During experiments the environmental chamber was operated in the constant-volume ("semi-batch") mode, in which clean air (11-14 lpm) was constantly added to make up for instrument sample flow. *

**16. Are the temperature and RH being constant throughout the experiment? When turning the light ON in general T and RH change due to reactions and heat from the UV lamps? Comment here. Time series of T and RH should be provided in Figure S1 top.**

The chamber conditions were controlled at 20℃ and 2% relative humidity (P3 L12).

We update Figure S1 by adding an additional panel with temperature and relative humidity (panel c):

[Figure]

*Figure S2 : (a) OH exposure,  (b) concentrations of O₃, NO₂, HONO+NOₓ, and (c) temperature and RH for a typical photooxidation experiment. NO₂ concentration was estimated using F0AM (Wolfe et al., 2016) based on MCM v3.3.1 (Bloss et al., 2005).*

**17. Page 3, line 13-14. "We performed a series of photochemical experiments, in which toluene and 1,2,4-TMB were oxidized by OH under high-NO conditions (Table S1)." Please provide NOₓ (NO and NO₂) concentrations in Table S1. What the authors refers to high NOₓ conditions? Provide references here!**

In our experiments the sum of the concentrations of nitrogen oxides and HONO was measured by 42i NOₓ monitor, and we modify the manuscript to make this clearer (P3 L29):

*The concentration  ozone (2B Technologies), relative humidity, and temperature were measured in the chamber. A 42i NOₓ monitor (Thermo Fischer Scientific) was used to measure the sum of concentrations of HONO and nitrogen oxides (NOₓ).*

Since individual levels of NO and NO₂ were not measured, they cannot be provided in Table S1. However, nitrogen oxides were primarily produced from HONO photolysis, and concentrations and time stamps of additional HONO injections are given in Table S1.

In this work we follow the definition of the high-NO chemistry regimes introduced by Seinfeld and Pandis (2016) (P3 L13):

*We performed a series of photochemical experiments, in which toluene and 1,2,4-TMB were oxidized by OH under high-NO conditions. The experiments were carried out under high-NO conditions such that the fate of peroxy radicals was primarily determined by their reaction with NO (Seinfeld and Pandis, 2016).*

At sufficiently high NO levels, the primary chain terminating reaction for peroxy radicals is the $RO_2 + NO$. This condition is called high-NO. On the contrary, if the fate of peroxy radicals is not determined by their reaction with NO, the condition is called low-NO. Under our experimental conditions, the $RO_2 + NO$ reaction is the primary loss of bicyclic peroxy radicals (Figs. S3 and S4). Hence, these experiments are called high-NO. At the same time, $NO_2$ levels observed in the environmental chamber are relevant for urban conditions as $NO_2$ concentration in major European cities varies from 3-38 ppb (Hazenkamp-von Arx et al., 2004).

**18. Please provide range of RH and T in Table S1 since term "approximative" was used.**

During photooxidation experiments, the chamber conditions were controlled at 20℃ and 2% relative humidity. However, during experiments temperature and relative humidity slightly changed as shown in Fig. S1.

In response to the comment, we change the following sentence (P3 L12):

*The temperature of the chamber was controlled at 292 (+/- 1) K. and approximately 2% relative humidity. All experiments were carried out under dry conditions (relative humidity, RH ≅2%, +/-1%) to simplify gas- and particle-phase measurements. Higher RH can potentially shorten the lifetime of particle-phase hydroperoxides, epoxides, and organonitrates (Li et al., 2018) as well as affect gas-particle partitioning kinetics and thermodynamics (Saukko et al., 2012).*

**19. Page 3, line 14. "First, dry ammonium sulfate particles, used as condensation nuclei, were injected in the chamber reach a number concentration of $(2.5 - 5.7) \cdot 10^4$ cm$^{-3}$." Table S1, shows particle loading after seed injection in the order of $(2.6 - 5.7) \cdot 10^4$ cm$^{-3}$ (no changes of the particle number after the light was ON and within the experimental error). Is this number stays the same throughout the experiment? Please clarify this in the footnote. This is important since HOMs were observed and will indicate if nucleation occurred or just growth of seed particles!! I'm curious how much aerosol mass was formed? - Table S1 should provide the amount of aerosol formed minus the seed aerosol injected. Specify the time of the reaction for these values? It is not stated in the manuscript that these values are associated with peak aerosol and compounds? - How much toluene was reacted? Since these experiments were conducted at low HC and NO$_x$ concentrations, uncertainties should be discussed, and error bars should be provided for toluene and 124TMB. Providing time series in figure S1 top for example or in a separate figure will make the manuscript stronger?**

The data is Table S1 corresponds to particle loading before the beginning of the experiment (i.e., before the lights were turned on). We add a footnote to Table S1 as suggested:

*Particle loading was measured before the chamber lights were turned on.*

In experiments #1-3 and 5-7, in which seed particles were injected, the total particle load decreased with time while organics to sulfate ratio increased (Fig. R2). Hence, low-volatile oxidation products produced in the chamber were primarily condensing on seed particles.

[Figure]

Figure R2: (a) Total particle loading measured by SMPS, and (b) organics to sulfate ratio measured by AMS in a 1,2,4-TMB photooxidation experiment.

On the contrary, in the experiments #4 and 8, in which no seed particles were injected, the total particle loading increased during the experiments (Fig. R3). This is an indication of nucleation happening in the chamber.

[Figure]

Figure R3: Total particle loading measured by SMPS in toluene photooxidation experiment #4, in which no seed particles were injected.

Total particle organics formed during experiments in this study was between 1.5-2 $\mu g/m^3$ as shown in Fig. 6.

In toluene experiments, 38-41 ppb of toluene was reacted over 5-6 hours of photooxidation. Uncertainties of the used instrumentation, including CIMS and PTR-MS instruments, are discussed in Section 2.2 (P4 L14-16).

**20. Page 3, line 16-17. "Nitrous acid (HONO) was later injected as an OH precursor." It is not clear here! Table S1 shows that HONO was injected initially and during the experiment. Please clarify this statement? Please provide the source of NO in this section?**

We change the following sentence (P3 L16):

*Next, nitrous acid (HONO) was  injected as an OH precursor.*

Additional aliquots of HONO were added to the chamber during experiments as stated in the manuscript (P3 L24).

We add the clarification on the source of NO (P3 L22):

*The reagents were allowed to mix for several minutes, after which the ultraviolet (UV) lights, centered ~340 nm, were turned on to start photolysis of HONO (resulting in the production of hydroxyl radicals and nitric oxide) and photooxidation of the precursor.*

**21. Page 3, line 18-19. Data in Table S1 not consistent with the statement here? HONO in Table S1 was between 28 and 60 ppbv. Either Table S1 or the text needs to be corrected?**

We change the sentence as suggested (P3 L18):

*15 lpm of subsequently injected purified air carried HONO into the chamber, which resulted in a mixing ratio  of 28-35  ppbv (except experiment 8 in which the initial HONO mixing ratio was 60 ppbv).*

**22. HONO was injected before the light was on. There is no background OH before the light was ON? Was the chamber in the dark before t=0? Figure S1 bottom shows background for $NO_2$, $O_3$, and HONO+$NO_x$? It is beneficial if $C_6F_6$ was presented also in Figure S1 top (dilution). - $O_3$ was high (~15 ppb) before light was ON! From where $O_3$ was generated before light ON?**

The time stamp t=0 on Fig. S1 corresponds to the moment when the chamber lights were turned on. Hence, the lights were turned off before that moment. HONO photolysis, which results in production of OH and NO, began when the lights were turned on. Therefore, there was no source of OH in the dark chamber, and the background OH levels were negligible.

During experiments, the constant flow of clean air was added to the chamber, and hexafluorobenzene was used to monitor dilution (P3 L11). Hence, we think it would be overwhelming to include hexafluorobenzene tracer in Fig. S1. However, in response to the comment, we include it here (Fig. R4).

[Figure]

Figure R4: Concentration of hexafluorobenzene used as a dilution tracer in a typical photooxidation experiment.

We also include the total dilution over the course of each experiments in Table S1 and add the following sentence in the manuscript (P3 L16):

*The total dilution over the course of experiments was 0.45-0.75 (except experiment 8 for which it was 0.27, Table S1).*

Ozone background levels could be explained by the chamber contamination and the offset of the ozone monitor. In this paper, we aimed to study oxidation of aromatic compounds under relevant urban conditions, not ozone-free conditions. If the background ozone contributed to photochemistry during experiments, we can expect it to do the same thing in the real atmosphere as well. Hence, we did not aim to conduct experiments under ozone-free conditions.

**23. "The concentration of NO in the chamber was estimated to be ~0.3 ppbv while NO$_2$ concentration was approximately 10 ppbv" I believe this is the initial concentration? In the manuscript the authors describe the experiments were conducted under high-NO? This is confusing? From this statement and Figure S1, the experiments were conducted under high NO$_2$? Please clarify this? How NO was estimated?**

We include the clarification that the NO$_x$ monitor was used to measure the sum of concentrations of NO$_x$ and HONO (P3 L29):

*The concentrations of  ozone (2B Technologies), relative humidity, and temperature were measured in the chamber. A 42i NO$_x$ monitor (Thermo Fischer Scientific) was used to measure the sum of concentrations of HONO and NO$_x$.*

We did not measure individual concentrations of NO and NO$_2$ and used F0AM setup based on MCM v3.3.1 to estimate levels of these compounds. The concentrations of NO and NO$_2$ stated in the manuscript (0.3 and 10 ppb, respectively) correspond to the average values during experiments (Fig. R5).

[Figure]

Figure R5: Estimated concentrations of NO and $NO_2$ during 1,2,4-TMB photooxidation experiment. Concentrations are estimated using F0AM setup based MCM v3.3.1.

In response to the comment we move the sentence towards the end of the section (P3 L19):

*~~The concentration of NO in the chamber was estimated to be ~0.3 ppbv while NO₂ concentration was approximately 10 ppbv~~. After the addition of the oxidant, the aromatic precursor (toluene or 1,2,4-TMB, Sigma-Aldrich) was added to the chamber by injecting 3 μL of the precursor into a heated inlet. The initial concentration of the precursor was 89 ppbv in toluene experiments and 69 ppbv in 1,2,4-TMB experiments. The reagents were allowed to mix for several minutes, after which the ultraviolet (UV) lights, centred ~340 nm, were turned on to start photolysis of HONO and photooxidation of the precursor. During experiments, additional injections of HONO were added to the chamber in order to roughly maintain the OH levels. As a result, the mixing ratio of NO in the chamber during experiments was estimated to be ~0.3 ppbv while the NO₂ mixing ratio was approximately 10 ppbv (Fig. S2).*

**24. I suggest adding experiment ID in Table S1. Then, associate the experiment ID in each figure/table and where the data is discussed in the text? It is hard to link the origin of the data when discussed in the text figures and tables?**

We add experiment IDs to Table S1 as suggested:

*Table S1: Description of experiments.*

| Expt. no. | VOC | Initial VOC concentration, ppbv | Initial HONO injection, ppbv | Additional HONO injections[a], ppbv | Particle loading after seed injection, cm$^{-3}$ | Temp., K | RH, % | Total dilution |
|---|---|---|---|---|---|---|---|---|
| 1 | toluene | 89 | 28 | 16 (124); 28 (220) | 2.6·10$^4$ | 292 | 2% | 0.44 |

Table S1: Description of experiments.

| Expt. no. | VOC | Initial VOC concentration, ppbv | Initial HONO injection, ppbv | Additional HONO injections[a], ppbv | Particle loading after seed injection, cm$^{-3}$ | Temp., K | RH, % | Total dilution |
|---|---|---|---|---|---|---|---|---|
| 1 | toluene | 89 | 28 | 16 (124); 28 (220) | $2.6 \cdot 10^4$ | 292 | 2% | 0.44 |
| 2 | toluene | 89 | 30 | 17 (140); 6 (180); 12 (290) | $4.2 \cdot 10^4$ | 292 | 2% | 0.62 |
| 3 | toluene | 89 | 31 | 21 (125); 18 (290) | $3.2 \cdot 10^4$ | 292 | 2% | 0.65 |
| 4 | toluene | 89 | 28 | 23 (180) | 100[b] | 292 | 3% | 0.45 |
| 5 | 124-TMB | 69 | 30 | 5 (155); 10 (300) | $3.6 \cdot 10^4$ | 292 | 2% | 0.56 |
| 6 | 124-TMB | 69 | 31 | 13 (145); 10 (245); 5 (345) | $3.8 \cdot 10^4$ | 292 | 2% | 0.72 |
| 7 | 124-TMB | 69 | 34 | 18 (150); 6 (265); 13 (390) | $5.7 \cdot 10^4$ | 292 | 2% | 0.74 |
| 8 | 124-TMB | 69 | 60 | - | 100[b] | 292 | 2% | 0.27 |

[a] The following format is used: HONO injection in ppbv (time since the beginning of the experiment in min).
[b] Particle loading was measured before the chamber lights were turned on.

**25. I'm confused with the term initial in Table S1. Are the authors referring before the light was ON? The system seems to be dynamic (clean air was continuously injected: dilution always occurring)" The initial concentrations of toluene and 1,2,4-TMB is very difficult to measure since the system was dynamic? I believe any quantitative data will be associated with high uncertainty? Provide error discussion in the manuscript! When conducting chamber experiments at low HC and NO$_x$ concentrations, initial HCs and NO$_x$ should be presented prior to light is ON and during the experiment. The instruments used should provide these data? Dilution data (C$_6$F$_6$) also should be provided in Figure S1?**

At the beginning of each experiment (before the chamber lights were turned on), nitrous acid and aromatic precursor were injected in the chamber (P3 L16-21). The initial concentration of aromatic precursor is well-constrained since the amount of the precursor (3 $\mu$l) injected in the chamber, the chamber volume, and the rate of dilution were known. Vocus-2R-PTR was directly calibrated for toluene and 1,2,4-TMB, and the initial concentration measured by the instrument agreed well with the calculated value. Instrumentational uncertainties are discussed in Section 2.2 of the manuscript (P4 L14-16).

We include the total dilution over the course of each experiment in Table S1:

| 2 | toluene | 89 | 30 | 17 (140); 6 (180); 12 (290) | $4.2 \cdot 10^4$ | 292 | 2% | 0.62 |
| 3 | toluene | 89 | 31 | 21 (125); 18 (290) | $3.2 \cdot 10^4$ | 292 | 2% | 0.65 |
| 4 | toluene | 89 | 28 | 23 (180) | $100^b$ | 292 | 3% | 0.45 |
| 5 | 124-TMB | 69 | 30 | 5 (155); 10 (300) | $3.6 \cdot 10^4$ | 292 | 2% | 0.56 |
| 6 | 124-TMB | 69 | 31 | 13 (145); 10 (245); 5 (345) | $3.8 \cdot 10^4$ | 292 | 2% | 0.72 |
| 7 | 124-TMB | 69 | 34 | 18 (150); 6 (265); 13 (390) | $5.7 \cdot 10^4$ | 292 | 2% | 0.74 |
| 8 | 124-TMB | 69 | 60 | - | $100^b$ | 292 | 2% | 0.27 |

$^a$ The following format is used: HONO injection in ppbv (time since the beginning of the experiment in min).
$^b$ Particle loading was measured before the chamber lights were turned on.

We also include the following sentence in the manuscript (P3 L16):

*The total dilution over the course of experiments was 0.45-0.75 (except experiment 8 for which it was 0.27, Table S1).*

**26. Page 3, line 22-23. "The reagents were allowed to mix for several min, …" It is not clear if the chamber was mixed well (since no fans were used) before the start of the reaction (light ON?) It is not easy to conduct these kings of experiments: the chamber is always under dilution "continuously with clean air" at the same time injecting a known amount of reagents before the light is ON. It is necessary to provide the time series of all parameters measured and used to conduct the experiment? T, RH, HONO, HCs, SOA formed, $C_6F_6$… [Note, the amount of HC injected and the volume of the chamber does not provide accurate concentrations!] In general, HCs and HONO as well other reagents should be injected continuously in the chamber until a stable concentration is attained for all reagents, then stop the HC injection and turn ON the light (start of the reaction). Comments from the authors is need here! I do see the use of $C_6F_6$ to measure the dilution rate and it was discussed rarely in this paper.**

The chamber mixing time can be calculated based on the aromatic precursor measured by Vocus-2R-PTR. After the precursor injection, we waited for a few minutes until the observed precursor level stabilized before turning the chamber lights on and starting the chemistry (P3 L22). Although no fans were used, we did not need them to keep the chamber well mixed: 11-14 slpm of clean air were constantly pushed into the chamber while the instruments were pulling ~7 slpm of air from the various points around the chamber, which resulted in turbulent mixing inside the chamber. In addition, mixing was promoted by fans external to the chamber blowing at the Teflon walls. As a result, the mixing time in the chamber was on the order of several minutes, which is much shorter than the duration of photooxidation experiments (several hours).

Time tracers of temperature, relative humidity, as well as concentrations of HONO+$NO_x$ and $O_3$ are shown in Fig. S1. In addition, SOA formed in toluene and 1,2,4-TMB experiments is shown in Fig. 6. As we

discussed in our response to comment 22, we think it would be overwhelming to include the hexafluorobenzene tracer in Fig. S1, but we have included it in this document (Fig. R4).

The environmental chamber was operated in the constant-volume ("semi-batch") mode. In response to the reviewer's comment, we have edited the following sentence (P3 L11):

*During experiments the environmental chamber was operated in the constant-volume ("semi-batch") mode, in which clean air (11-14 lpm) was constantly added to make up for instrument sample flow. clean air was continuously added to the chamber to keep its volume constant.*

**27. Table S1. Particle loading should be (2.6 - 5.7) $10^4$ instead of (2.6 - 3.5)$10^4$ as stated in line 15, page 1.**

P3 L14-15 of the manuscript read:

*First, dry ammonium sulfate particles were injected in the chamber to reach a number concentration of $2.5$-$5.7 \cdot 10^4$ $cm^{-3}$*

**28. Suggest using "Experimental section" instead of "Experimental design"**

In our opinion, Section 2.1 provides the reader with the general description of conducted experiments (i.e., sequence in which various components were added to the chamber and operating conditions). As such, we think that the current title of this Section better describes its content.

**29. Pages 3-4: chamber instrumentation. The description of the instruments is difficult to follow and switching between CIMS and PTR notations is hard to follow. All instruments use chemical ionization. At the beginning the authors state that four instruments were used (two CIMS and two PTR) and in this section it seems that three instruments were used: (1) page 4, line 1 ("including the $I^-$ CIMS instrument; and two PTR (CIMS $H_3O^+$ and $NH_4^+$. This section needs to be clarified and consistent with the manuscript. Recently artefacts and sampling issues were reported, and this should be discussed in the paper (see my general comments).**

We change the description of used instrumentation in Introduction (P3 L4):

*We use four high-resolution time-of-flight chemical ionization mass spectrometers ($NH_4^+$ CIMS, $I^-$ CIMS and two PTR-MS) to characterize and quantify gas- and particle-phase oxidation products. We use three chemical ionization mass spectrometry (CIMS) techniques ($I^-$ reagent ion, $NH_4^+$ reagent ion, and $H_3O^+$ reagent ion) to characterize and quantify gas-phase oxidation products. In addition, $NH_4^+$ CIMS and $I^-$ CIMS were used to detect particle-phase products.*

We also change Section 2.2 (P3 L30):

*Aromatic precursors as well as gas-phase oxygenated volatile organic compounds (OVOCs) were detected by chemical ionization high-resolution time-of-flight mass spectrometry (CIMS) instruments, including the $I^-$ CIMS instrument (Aerodyne Research Inc.; Lee et al., 2014) and two proton-transfer-reaction mass-spectrometry (PTR-MS) instruments: Vocus-2R-PTR-TOF (TOFWERK A.G.; Krechmer et al., 2018) and PTR3 (Ionicon Analytik; Breitenlechner et al., 2017). The latter instrument was operated in a switching-mode regime using $H_3O^+ \cdot (H_2O)_n$, n=0-1 (as $H_3O^+$-CIMS) and $NH_4^+ \cdot (H_2O)_n$, n = 0-2 (as $NH_4^+$-CIMS) primary ions (Hansel et al., 2018; Zaytsev et al., 2019). Switching between ion modes occurred every five minutes.*
*Aromatic precursors as well as gas-phase oxygenated volatile organic compounds (OVOCs) were detected*

*by chemical ionization high-resolution time-of-flight mass spectrometry (CIMS) techniques: I⁻ CIMS, NH₄⁺ CIMS, and H₃O⁺ CIMS. The I⁻ CIMS instrument (Aerodyne Research Inc.) is described by Lee et al. (2014). NH₄⁺ CIMS and H₃O⁺ CIMS were carried out using a mass spectrometer (PTR3, Ionicon Analytik) which was operated in two ionization modes: $H_3O^+\cdot(H_2O)_n$, n=0-1 (as PTR3 H₃O⁺ CIMS, Breitenlechner et al., 2017) and $NH_4^+\cdot(H_2O)_n$, n=0-2 (as PTR3 NH₄⁺ CIMS, Zaytsev et al., 2019). Switching between ion modes occurred every five minutes. H₃O⁺ CIMS was also conducted by a proton-transfer-reaction mass spectrometer (Vocus-2R-PTR, Aerodyne Research Inc.; Krechmer et al., 2018).*

**30. What reagent gas was used for the Vocus-2R-PTR?**

Vocus-2R-PTR uses hydronium ($H_3O^+$) ions to ionize VOCs (Krechmer et al., 2018).

**31. The authors should provide the 10 compounds used in the calibration of I⁻CIMS and PTR3 in SI. Using these instruments for aerosol characterization may be associated with sampling artefacts as discussed recently (see my general comments)**

We include the following tables containing calibrated species and measured sensitivities in the SI:

*Table S2: Sensitivities of I⁻ CIMS for calibrated species.*

| Species | Ion formula | m/z | Sensitivity (ndcps/ppb) |
|---|---|---|---|
| Formic acid | $CH_2O_2I^-$ | 172.91 | 3000 |
| Acetic acid | $C_2H_4O_2I^-$ | 186.93 | 130 |
| Acrylic acid | $C_3H_4O_2I^-$ | 198.93 | 63 |
| Glycolic acid | $C_2H_4O_3I^-$ | 202.92 | 880 |
| cis-2-butene-1,4-diol | $C_4H_8O_2I^-$ | 214.96 | 1600 |
| 1,2-butanediol | $C_4H_{10}O_2I^-$ | 216.97 | 110 |
| Phenol | $C_6H_6OI^-$ | 220.95 | 180 |
| Malonic acid | $C_3H_4O_4I^-$ | 230.92 | 74 |
| o-cresol | $C_7H_8OI^-$ | 234.96 | 65 |
| Nitrophenol | $C_6H_5NO_3I^-$ | 265.93 | 38500 |

*Table S3: Sensitivities of PTR3 H₃O⁺ CIMS and PTR3 NH₄⁺ CIMS for calibrated species.*

| Species | PTR3 H₃O⁺ CIMS | | | PTR3 NH₄⁺ CIMS | | |
|---|---|---|---|---|---|---|
| | Ion formula | m/z | Sensitivity (ndcps/ppb) | Ion formula | m/z | Sensitivity (ndcps/ppb) |
| Acetone | $C_3H_6OH^+$ | 59.05 | 12900 | $C_3H_6ONH_4^+$ | 76.08 | 10600 |
| Acetic acid | $C_2H_4O_2H^+$ | 61.03 | 9600 | $C_2H_4O_2NH_4^+$ | 78.06 | 840 |
| Methacrolein | $C_4H_6OH^+$ | 71.05 | 15300 | $C_4H_6ONH_4^+$ | 88.08 | 4900 |
| 2-furanone | $C_4H_4O_2H^+$ | 85.03 | 20700 | $C_4H_4O_2NH_4^+$ | 102.06 | 10300 |
| Diacetyl | $C_4H_6O_2H^+$ | 87.04 | 5000 | $C_4H_6O_2NH_4^+$ | 104.07 | 5200 |
| Angelica lactone | $C_5H_6O_2H^+$ | 99.04 | 19400 | $C_5H_6O_2NH_4^+$ | 116.07 | 20800 |
| Benzaldehyde | $C_7H_6OH^+$ | 107.05 | 16200 | $C_7H_6ONH_4^+$ | 124.07 | 14500 |
| o-cresol | $C_7H_8OH^+$ | 109.07 | 6900 | $C_7H_8ONH_4^+$ | 126.09 | 370 |
| 1,2,4-TMB | $C_9H_{12}H^+$ | 121.10 | 1640 | $C_9H_{12}NH_4^+$ | 138.13 | 800 |
| 3-decanone | $C_{10}H_{20}OH^+$ | 157.16 | 13600 | $C_{10}H_{20}ONH_4^+$ | 174.19 | 23200 |

**32. Page 5, lines 4-24. Please provide time series for [ArVOC]$_0$ for toluene and 124-TMB as I suggested before? Data obtained from PTR!**

Time series of toluene and 1,2,4-TMB levels measured by Vocus-2R-PTR and dilution-corrected concentrations are shown in Fig. R6.

[Figure]

Figure R6: Dilution corrected and not dilution corrected concentrations of (a) toluene and (b) 1,2,4-TMB in photooxidation experiments.

**33. Page 5. The yields were described but were not provided in the paper!**

The yields of major first-generation oxidation products for both toluene and 1,2,4-TMB oxidation experiments are given in Tables 1 and 2.

**34. What "b" represents in eq. 3?**

Parameter $b$ is the offset. Since it does not have any chemical sense, we leave it out of the equation:

$$[X]^{corr} = Y[ArVOC]^{reacted} \; \cancel{+\,b} \tag{3}$$

**35. Page 4, line 31. Is the light intensity was measured?**

The spectrum of the chamber lights was measured, while the total light intensity is unknown. Hence, the chamber light intensity in the model was tuned to match the measured time-dependent concentrations of aromatic compounds with the modelled values (P4 L31).

**36. Page 4, 5: Gamma Kinetic parameterization. Can the structure affect the results (the parameters obtained)? This model is applied to formulas here?**

GKP was applied to the oxidation products observed in this study. The generation parameter $m$ and kinetic parameter $k$ for a given species depend on the number of reactions with OH needed to produce that species and the effective second-order rate constant, respectively. Therefore, compounds with different structures will likely have different parameters $m$ and $k$. However, CIMS and PTR-MS instruments cannot distinguish between isomers and do not provide further insight into the molecular structure of detected compounds. Hence, it is possible that several compounds with the same molecular formulas, but different

structures were lumped together and returned a non-integer parameter *m*. The generation parameter *m* and kinetic parameter *k* for major oxidation products are given in Tables 1 and 2.

**37. Page 6, line 18. Figure 1 should incorporate both HCs as I stated before.**

We include Figure 1 in this paper to illustrate four major oxidation pathways common for numerous aromatic compounds (including toluene and 1,2,4-TMB). Hence, we think it would be overwhelming to add 1,2,4-TMB oxidation scheme to this figure as toluene and 1,2,4-TMB schemes are quite similar. However, in response to this comment we modify the following sentence (P6 L17):

* These four channels are illustrated using the example of toluene in Fig. 1.*

**38. Page 10, line 18. Define TD-NH$_4^+$ CIMS.**

The definition of TD-NH$_4^+$ CIMS was given on page 4, line 24 (NH$_4^+$ CIMS instrument equipped with thermal denuder).

**39. Figure 6. Figure 6 title. I think there is an error! (a) should be toluene and (b) should be 124-TMB. Please clarify? Also is total organics or total organics carbon?**

We thank the reviewer for spotting this typo and update the title as suggested:

*Figure 6: Total organics measured in the particle phase by NH$_4^+$ CIMS and binned by the carbon atom number (a) in the toluene  photooxidation experiment and (b) the 1,2,4-TMB  photooxidation experiment. Total carbon measured by AMS is in red.*

**40. Page 10. "The O:C ratios calculated from individual species measured from thermally desorbed SOA using NH$_4^+$ CIMS were ~0.95 for toluene SOA and ~0.7 for 1,2,4-TMB SOA. These ratios are in good agreement with the atomic O:C ratios measured by AMS (0.85 and 0.65 for toluene and 1,2,4-TMB SOA, respectively) (Canagaratna et al., 2015)." Does this comparison was applied to the same SOA size for the AMS and CIMS?**

AMS transmission efficiency is maximum (100%) for particles with diameters from 100 nm to 550 nm (Knote et al., 2015). PTR3 instrument used for the particle-phase measurements was equipped with a gas-phase denuder (Ionicon Analytic GmbH, Austria) that removes the gas-phase organics. Particles smaller than 100 nm in diameter do not pass through a denuder and are not analysed in the instrument (Eichler et al., 2015). Hence, the AMS and CIMS instruments were used to analyze the composition of aerosols of the same size.

**41. Page 10. "Products observed in the gas phase are compared to those detected in the particle phase to further understand the mechanism of SOA formation from aromatics precursors." Can partitioning coefficients be obtained for these products? Please provide organic species present in both gas phase and particle phase and their estimated partitioning coefficient? Table for example.**

Partitioning coefficients for low-volatile compounds could be obtained if the equilibrium between the gas and particle-phase organics was achieved. However, there are a number of reasons why simple gas-particle partitioning might not explain the concentrations observed in the two phases. First, chamber lights were turned on during the whole duration of experiments resulting in constant production and loss of oxidation products in the gas phase. Second, particles were lost on the chamber walls, while the wall

loss term was not estimated. Third, CIMS and PTR-MS instruments cannot distinguish between isomers and do not provide further insight into the molecular structure of detected compounds. Hence, it is possible that several compounds with the same molecular formulas, but different structures and partitioning coefficients were lumped together. Therefore, partitioning coefficients for observed species cannot be reported.

**42. The section describing on page 10, lines 25 - 32. "non-fragmentary" "ring-retaining" "ring scission" "fragmentary" is not clear to me. Could the authors describe how the thermal fragmentation either in the gas phase or the particle phase reported on page 10 can affect the data (formula and structures) provided in this manuscript? Are the HOMs detected not originated from artefacts in the inlets? These compounds with high O:C ratio are expected to be in the particle phase but numerous studies including this one report them in the gas phase?**

During particle-phase measurements, sampled air passes through a gas-phase denuder (Ionicon Analytik GmbH, Austria) that removes the gas-phase organics and then through a thermal desorption region heated to 180℃ that vaporizes the aerosol particles. This may result in thermal decomposition of OVOCs. We studied thermal decomposition of OVOCs extracted from alpha-pinene SOA by measuring their peak intensities using TD-NH$_4^+$ CIMS. Signals of many species increased at moderate temperatures ($T < 160$℃) and levelled out or decreased at higher temperatures ($T > 180$℃), as shown in Fig. R1 (Zaytsev et al., 2019). Therefore, we chose the TDU temperature to be 180℃, as at this temperature the major fraction of particles was evaporated while thermal decomposition of labile species was relatively small.

We modify the description of TD-NH$_4^+$ CIMS and TD-H$_3$O$^+$ CIMS (P4 L21):

*Particle-phase compounds were quantified using the FIGAERO-HRToF-I$^-$ CIMS (Lopez-Hilfiker et al., 2014), and a second PTR3 that could be operated in two positive modes as described above and equipped with an aerosol inlet comprising a gas-phase denuder and a thermal desorption unit heated to 180℃ (TD-NH$_4^+$ CIMS and TD-H$_3$O$^+$ CIMS). At this temperature, all particles were evaporated while thermal decomposition of labile oxygenated compounds was relatively small (Zaytsev et al., 2019).*

Low volatile organic compounds (LVOC) including HOMs have low saturation vapour pressure such that almost every collision with wall inlet leads to a complete loss. However, the estimates for these losses in the literature have shown significant discrepancy. Breitenlechner et al. (2017) estimated the wall losses for LVOC with more than five oxygens in the PTR3 inlet to be 80% while for VOC with less than five oxygens the wall losses were assumed to be negligible. Hansel et al. (2018) evaluated the wall losses in the CI3-ToF inlet to be 50%. Since we did not have an additional instrument with calibrated diffusion losses in the inlet (i.e., acetate CIMS), we did not take into account wall losses of less volatile species in inlets of used instrumentation. It results in underestimation of the yield of these compounds including HOMs.

In order to avoid potential ambiguity, we replace the term "non-fragmentary" by "non-fragmented" throughout the manuscript.

**43. In general, chamber backgrounds NO$_x$ always is present (although clean air was used) due to wall chemistry and the history of the chamber (heterogeneous wall chemistry). Comments!**

Between experiments, the environmental chamber was cleaned by flushing with purified air for at least 10 hours while the chamber lights were turned on. As a result, NO$_x$ background levels before experiments

were less than 5 ppbv (Fig. S1). However, HONO+NO$_x$ levels during experiments were much higher (~30 ppbv, Fig. S1).

**References:**

Breitenlechner, M., Fischer, M., Hainer, M., Heinritzi, M., Curtius, M., and Hansel, A.: PTR3: An instrument for Studying the Lifecycle of Reactive Organic Carbon in the Atmosphere, Anal. Chem., 89, 5824–5831, https://doi.org/10.1021/acs.analchem.6b05110, 2017.

Buhr, K., van Ruth, S., and Delahunty, C.: Analysis of volatile flavour compounds by Proton Transfer Reaction-Mass Spectrometry: fragmentation patterns and discrimination between isobaric and isomeric compounds, Int. J. Mass Spectrom., 221, 1–7, https://doi.org/10.1016/S1387-3806(02)00896-5, 2002.

Calvert, J., Atkinson, R., Becker, K.H., Kamens, R., Seinfeld, J., Wallington, T., and Yarwood, G.: The mechanisms of atmospheric oxidation of aromatic hydrocarbons, Oxford University Press, Inc., New York, 2002.

Eichler, P., Müller, M., D'Anna, B., and Wisthaler, A.: A novel inlet system for online chemical analysis of semi-volatile submicron particulate matter, Atmos. Meas. Tech., 8, 1353–1360, https://doi.org/10.5194/amt-8-1353-2015, 2015.

Erickson, M. H., Gueneron, M., and Jobson, B. T.: Measuring long chain alkanes in diesel engine exhaust by thermal desorption PTR-MS, Atmos. Meas. Tech., 7, 225–239, https://doi.org/10.5194/amt-7-225-2014, 2014.

Hansel, A., Scholz, W., Mentler, B., Fischer L., and Berndt, T.: Detection of RO2 radicals and other products from cyclohexene ozonolysis with NH4+ and acetate ionization mass spectrometry, Atmos. Env., 186, 248–255, doi:10.1016/j.atmosenv.2018.04.023, 2018.

Hazenkamp-von Arx, M.E., Gotschi, T., Ackermann-Liebrich, U., Bono, R., Burney, P., Cyrys, J., Jarvis, D., Lillienberg, L., Luczynska, C., Maldonado J.A., Jaen, A., de Marco, R., Mi, Y., Modig L., Bayer-Oglesby, L., Payo, F., Soon, A., Sunyer, J., Villani, S., Weyler, J., and Kunzli N.: PM$_{2.5}$ and NO$_2$ assessment in 21 European study centres of ECRHS II: annual means and seasonal differences, Atmos. Env., 38, 1943–1953, https://doi.org/10.1016/j.atmosenv.2004.01.016, 2004.

Jang, M., and Kamens, R.M.: Characterization of Secondary Aerosol from the Photooxidation of Toluene in the Presence of NO$_x$ and 1-Propene, Environ. Sci. Technol., 35, 3626-3639, https://doi.org/10.1021/es010676+, 2001.

Kleindienst, T.E., Conver, T.S., McIver, C.D., and Edney, E.O.: Determination of Secondary Organic Aerosol Products from the Photooxidation of Toluene and their Implications in Ambient PM$_{2.5}$, Journal of Atmospheric Chemistry, 47, 79-100, https://doi.org/10.1023/B:JOCH.0000012305.94498.28, 2004.

Knote, C., Brunner, D., Vogel, H., Allan, J., Asmi, A., Äijälä, M., Carbone, S., van der Gon, H. D., Jimenez, J. L., Kiendler-Scharr, A., Mohr, C., Poulain, L., Prévôt, A. S. H., Swietlicki, E., and Vogel, B.: Towards an online-coupled chemistry-climate model: evaluation of trace gases and aerosols in COSMO-ART, Geosci. Model Dev., 4, 1077–1102, https://doi.org/10.5194/gmd-4-1077-2011, 2011.

Krechmer, J.E., Lopez-Hilfiker, F., Koss, A., Hutterli, M., Stoermer, C., Deming, B., Kimmel, J., Warneke, C., Holzinger, R., Jayne, J., Worsnop, D., Fuhrer, K., Gonin, M., and de Gouw J.: Evaluation of a New Vocus

Reagent-Ion Source and Focusing Ion-Molecule Reactor for use in Proton-Transfer-Reaction Mass Spectrometry, Anal. Chem., 90, 20, 12011-12018, doi:10.1021/acs.analchem.8b02641, 2018.

Li, Z., Smith, K. A., and Cappa, C. D.: Influence of relative humidity on the heterogeneous oxidation of secondary organic aerosol, Atmos. Chem. Phys., 18, 14585–14608, https://doi.org/10.5194/acp-18-14585-2018, 2018.

Pagonis, D., Krechmer, J. E., de Gouw, J., Jimenez, J. L., and Ziemann, P. J.: Effects of gas–wall partitioning in Teflon tubing and instrumentation on time-resolved measurements of gas-phase organic compounds, Atmos. Meas. Tech., 10, 4687–4696, https://doi.org/10.5194/amt-10-4687-2017, 2017.

Saukko, E., Lambe, A. T., Massoli, P., Koop, T., Wright, J. P., Croasdale, D. R., Pedernera, D. A., Onasch, T. B., Laaksonen, A., Davidovits, P., Worsnop, D. R., and Virtanen, A.: Humidity-dependent phase state of SOA particles from biogenic and anthropogenic precursors, Atmos. Chem. Phys., 12, 7517-7529, https://doi.org/10.5194/acp-12-7517-2012, 2012.

Seinfeld, J.H., and Pandis, S.N.: Atmospheric Chemistry and Physics: From Air Pollution to Climate Change, Wiley, Hoboken NJ, 2016.

Yuan, B., Koss, A.R., Warneke, C., Coggon, M., Sekimoto, K., and de Gouw, J.A.: Proton-Transfer-Reaction Mass Spectrometry: Applications in Atmospheric Sciences, Chem. Rev., 117, 13187-13229, https://doi.org/10.1021/acs.chemrev.7b00325, 2017.

Wu, R., Pan, S., Li, Y., and Wang, L.: Atmospheric Oxidation Mechanism of Toluene, J. Phys. Chem. A., 118, 4533-4547, https://doi.org/10.1021/jp500077f, 2014.

Zaytsev, A., Breitenlechner, M., Koss, A. R., Lim, C. Y., Rowe, J. C., Kroll, J. H., and Keutsch, F. N.: Using collision-induced dissociation to constrain sensitivity of ammonia chemical ionization mass spectrometry (NH4+ CIMS) to oxygenated volatile organic compounds, Atmos. Meas. Tech., 12, 1861–1870, https://doi.org/10.5194/amt-12-1861-2019, 2019.

---

## Author Comment (AC2) · 5 Nov 2019

**Response to Reviewer 2**

Reviewer comments are in **bold**. Author responses are in plain text. Excerpts from the manuscript are in *italics*. Modifications to the manuscript are in *blue italics*. Page and line numbers in the responses correspond to those in the original ACPD paper.

**In this work, the authors presented results from oxidation experiments of aromatic compounds, toluene and 1,2,4-TMB. These aromatic compounds are important VOCs in urban areas, and their oxidation leads to significant ozone and secondary organic aerosol (SOA) formation. In this study the authors employed a number of new analytical techniques to measure the gas and particle phase composition, and compared to the latest version of Master Chemical Mechanism (MCM), which summarizes the current understanding about the mechanisms. Furthermore, the time trend analysis using gamma kinetic parameterization is a novel method to look at the multigenerational chemistry. This manuscript is well written, and I only have some minor suggestions. I recommend publication of this manuscript in ACP.**

We would like to thank the reviewer for the positive reception of our work and constructive comments that helped us to improve our manuscript. In this document we provide our replies to the reviewer's comments.

**1. It would be good to know on a bulk or general level, how these results improve the understanding of the chemistry. For example, I wonder what the carbon closure now is, with these new measurements. Figure 3 is probably a good place to show that.**

The objective of this study is to evaluate the importance of various gas-phase oxidation pathways of aromatic compounds in terms of production of oxygenated low-volatile species (including HOMs) and SOA formation potential (P3 L5). Bulk organic carbon properties such as volatility, oxidation state, and reactivity, as well as carbon closure will be discussed in an accompanying paper currently under preparation.

*2. Somewhat related: One key piece of information shown in Section 3.3 and Fig. 6 is that the total SOA mass measured by AMS and NH$_4^+$ CIMS compare very well, and so do the O/C ratios. This is an important discovery and should be highlighted in the abstract.*

We agree with the reviewer that this is an important discovery. We underline this finding in Conclusions (P11 L28):

*Many of these compounds are low in volatility and comprise a significant fraction (more than 25%) of SOA mass, which was measured using AMS and TD-NH$_4^+$ CIMS and the two measurements are in good agreement.*

*3. The multigenerational chemistry of many of the products is a key contribution. I expect that the accompanying paper describing the methods will be well received. There are some ambiguous ones that have non-integer m (e.g. 1.7-1.8). What is the general uncertainty in this analysis?*

Noise can contribute to uncertainty in returned values of *m*. At low generations (*m*=1-2), the standard deviation of the fit is 0.1, while it can be higher (up to 0.8) at higher (3+) generations. Hence, it is possible

that the compounds with non-integer $m$ are produced by more than one pathway with different number of reaction steps.

We add the following sentence to the manuscript (P6 L13):

*At low generations (m=1-2), the standard deviation of the fit is 0.1, while it can be higher (up to 0.8) at higher (3+) generations (Koss et al., 2019).*

**4. Related to comment/question 3: I expect that some experiments with oxidation of later generation products would be very helpful. For example, oxidation of cresol (which is commercially available) should yield lower m for some of the products. Perhaps even examining the decrease in m would help apportion the relative amount for each generation. I think these are important experiments anyway given that the authors are claiming the importance of phenolic and benzaldehyde pathways in HOM production.**

We agree with the reviewer that experiments with oxidation of later generation products (e.g., cresols and benzaldehydes) would be very helpful to further investigate different pathways in which highly oxygenated compounds are produced. While we plan to conduct these experiments in the future, we think that they are out of the scope of the current work.

**5. The experiments were all conducted under RH of 2%. While I completely understand the rationale to create a well-controlled environment, it may be worthwhile to mention this is a potential limitation of this study and discuss implications. I do not see water playing an important role in the gas-phase chemistry, but could potentially shorten the lifetime of particle-phase hydroperoxides, epoxides and organic nitrates.**

We add the following sentence as suggested (P3 L12):

*The temperature of the chamber was controlled at 292 (+/- 1) K . All experiments were carried out under dry conditions (relative humidity, RH $\cong$2%, +/- 1%) to simplify gas- and particle-phase measurements. Higher RH can potentially shorten the lifetime of particle-phase hydroperoxides, epoxides and organonitrates (Li et al., 2018) as well as affect gas-particle partitioning kinetics and thermodynamics (Saukko et al., 2012).*

**Methods: I do not understand why the authors would use hexafluorobenzene as a tracer for both chamber wall loss and dilution of VOCs. I can see hexafluorobenzene is a good tracer for dilution, but I do not expect it to be lost to the chamber walls. Based on the chamber volume and air refilling rate, the dilution rate can be estimated. Is the hexafluorobenzene decaying faster than this dilution rate? If so, why is it being lost to the walls?**

We used hexafluorobenzene as a tracer for the dilution of VOCs, not as a tracer for vapour-phase wall loss. The reviewer is correct that a highly volatile compound like hexafluorobenzene is not expected to be lost to the chamber walls. The hexafluorobenzene loss rate was consistent with the known chamber volume and dilution air flow. Vapour wall loss was quantified using the "rapid burst" method (Krechmer et al., 2016), and the rate constant of this process $k_{\text{wall loss}}$ was estimated to be $5 \cdot 10^{-4}$ s$^{-1}$.

To make this point clearer, we edit the following sentences:

P4 L32: *The  dilution term for volatile compounds was estimated based on the concentration of the dilution tracer, hexafluorobenzene.*

P5 L4: *The OH concentration was determined using the decay of the aromatic precursor, accounting for losses from dilution .*

We also update equation 1:

$$[\text{ArVOC}]_t = [\text{ArVOC}]_0 \cdot \exp\left(- k_{\text{ArVOC+OH}} \cdot [\text{OH}_{\text{exposure}}]_t - k_{\text{\st{physical loss ArVOC} dilution}} \cdot t \right) \qquad (1)$$

**CO and formaldehyde were mentioned in methods, but no results were presented.**

CO and formaldehyde will be discussed in further details in the accompanying paper under preparation. Since they play no role in the present paper, we remove the sentence describing CO and formaldehyde measurements from the manuscript (P4 L17).

**Methods: Particle-phase compounds were quantified using I⁻ CIMS, but for the gas phase compounds the authors claim I⁻ CIMS is quite uncertain. Are the uncertainties in quantification the same for both phases?**

Particle-phase compounds were quantified using both the FIGAERO-HRToF-I⁻-CIMS and a second PTR3 equipped with an aerosol inlet comprising a gas-phase denuder and a thermal desorption unit (TD-NH₄⁺ CIMS). Total organic mass and O:C ratio measured by TD-NH₄⁺ CIMS are in good agreement with the AMS measurements (P10 L21). As for FIGAERO-HRToF-I⁻-CIMS, uncertainties of the particle-phase measurements are similar to the gas-phase measurements.

We add the following clarification to the manuscript (P4 L24):

*Uncertainties of the particle-phase CIMS measurements are similar to the corresponding uncertainties of the gas-phase CIMS instruments.*

**Section 2.4: how large are the time steps?**

The time step $\Delta t$ was five minutes which corresponds to the switching period between two ionization modes of PTR3 (P4 L5).

**Section 3.1: Is it possible that the epoxide was not detected because of thermal decomposition for the particle phase measurements, or fragmentation during ionization?**

The reviewer raises an interesting point. While it is possible that the epoxide was decomposed or fragmented during ionization, we think that it should have been detected using the instrumentation implemented in this study. We calibrated PTR3 NH₄⁺ CIMS and H₃O⁺ CIMS for isoprene epoxydiol (trans-IEPOX) and observed very little fragmentation in NH₄⁺ CIMS comparing to H₃O⁺ CIMS (Fig. R1).

[Figure]

Figure R1: High-resolution mass-spectra obtained during calibration of trans-IEPOX in (a) PTR3 $NH_4^+$ CIMS and (b) PTR3 $H_3O^+$ CIMS.

In addition, we studied thermal decomposition of OVOCs extracted from alpha-pinene SOA by measuring their peak intensities using TD-$NH_4^+$ CIMS. Signals of many species increased at moderate temperatures ($T<160°C$) and levelled out or decreased at higher temperatures ($T>180°C$), as shown in Fig. R2. Therefore, we chose the TDU temperature to be 180°C, as at this temperature all particles were evaporated while thermal decomposition of labile species was relatively small.

[Figure]

Figure R2: Thermograms of select OVOCs extracted from alpha-pinene ozonolysis SOA.

Hence, we think that toluene and 1,2,4-TMB epoxides should be detectable using $NH_4^+$ CIMS.

**Section 3.2.1, Line 13: BPR has been defined earlier.**

We thank the reviewer for spotting this typo in the manuscript and remove the abbreviation ("BPR") from the manuscript (P7 L13).

**Section 3.2.1 Line 29-30: Presumably the lifetimes are calculated using generic $RO_2$+NO and $RO_2$+$HO_2$ rate constants? What rate constants were used?**

The rate constants were estimated using information from MCM v3.3.1.
For toluene:
$k_{RO_2+HO_2} = 2.05 \cdot 10^{-11}$ cm³ molecule$^{-1}$ s$^{-1}$, $k_{RO_2+NO} = 9.26 \cdot 10^{-12}$ cm³ molecule$^{-1}$ s$^{-1}$.
For 1,2,4-TMB:
$k_{RO_2+HO_2} = 2.22 \cdot 10^{-11}$ cm³ molecule$^{-1}$ s$^{-1}$, $k_{RO_2+NO} = 9.26 \cdot 10^{-12}$ cm³ molecule$^{-1}$ s$^{-1}$.

**Tables 1-3: what are the uncertainties in m from the fits?**

Noise can contribute to uncertainty in returned values of $m$.

We add the following sentence to the manuscript (P6 L13):

*At low generations (m=1-2), the standard deviation of the fit is 0.1, while it can be higher (up to 0.8) at higher (3+) generations (Koss et al., 2019).*

**Figure 6: why is there a discontinuity at 14 hours of exposure for toluene? Was more OH precursor added? Similarly there seems to be one as well for TMB at 4 h.**

During experiments, additional injections of HONO were added to the chamber to roughly maintain the OH levels (Fig. S1). Discontinuities in particle-phase signals are indeed caused by the additional HONO injections.

**References**

Krechmer, J.E., Pagonis, D., Ziemann, P.J., and Jimenez, J.L.: Quantification of Gas-Wall Partitioning in Teflon Environmental Chambers Using Rapid Bursts of Low-Volatility Oxidized Species Generated in Situ, Environ. Sci. Technol. 50(11), 5757–5765, https://doi.org/10.1021/acs.est.6b00606, 2016.

Li, Z., Smith, K. A., and Cappa, C. D.: Influence of relative humidity on the heterogeneous oxidation of secondary organic aerosol, Atmos. Chem. Phys., 18, 14585–14608, https://doi.org/10.5194/acp-18-14585-2018, 2018.

Saukko, E., Lambe, A. T., Massoli, P., Koop, T., Wright, J. P., Croasdale, D. R., Pedernera, D. A., Onasch, T. B., Laaksonen, A., Davidovits, P., Worsnop, D. R., and Virtanen, A.: Humidity-dependent phase state of SOA particles from biogenic and anthropogenic precursors, Atmos. Chem. Phys., 12, 7517-7529, https://doi.org/10.5194/acp-12-7517-2012, 2012.